# Def6 regulates endogenous type-I interferon responses in osteoblasts and suppresses osteogenesis

Zhonghao Deng[1†], Courtney Ng[1†], Kazuki Inoue[1,2†], Ziyu Chen[1], Yuhan Xia[1], Xiaoyu Hu[3], Matthew Greenblatt[4,5], Alessandra Pernis[6,7], Baohong Zhao[1,2,8]*

[1]Arthritis and Tissue Degeneration Program and David Z. Rosensweig Genomics Research Center, Hospital for Special Surgery, New York, United States; [2]Department of Medicine, Weill Cornell Medical College, New York, United States; [3]Institute for Immunology and School of Medicine, Tsinghua University, Beijing, China; [4]Pathology and Laboratory Medicine, Weill Cornell Medical College, New York, United States; [5]Research Division, Hospital for Special Surgery, New York, United States; [6]Autoimmunity and Inflammation Program, Hospital for Special Surgery, New York, United States; [7]Graduate Program in Immunology and Microbial Pathogenesis, Weill Cornell Graduate School of Medical Sciences, New York, United States; [8]Graduate Program in Cell and Development Biology, Weill Cornell Graduate School of Medical Sciences, New York, United States

*For correspondence:
zhaob@hss.edu

[†]These authors contributed equally to this work

Competing interests: The authors declare that no competing interests exist.

**Abstract** Bone remodeling involves a balance between bone resorption and formation. The mechanisms underlying bone remodeling are not well understood. DEF6 is recently identified as a novel loci associated with bone mineral density. However, it is unclear how Def6 impacts bone remodeling. We identify Def6 as a novel osteoblastic regulator that suppresses osteoblastogenesis and bone formation. Def6 deficiency enhances both bone resorption and osteogenesis. The enhanced bone resorption in Def6$^{-/-}$ mice dominates, leading to osteoporosis. Mechanistically, Def6 inhibits the differentiation of both osteoclasts and osteoblasts via a common mechanism through endogenous type-I IFN-mediated feedback inhibition. RNAseq analysis shows expression of a group of IFN stimulated genes (ISGs) during osteoblastogenesis. Furthermore, we found that Def6 is a key upstream regulator of IFNβ and ISG expression in osteoblasts. Collectively, our results identify a novel immunoregulatory function of Def6 in bone remodeling, and shed insights into the interaction between immune system and bone.

## Introduction

Healthy bone mass in adults is primarily maintained by constant and dynamic remodeling between osteoclast-mediated bone resorption and osteoblast/osteocyte-mediated bone formation (*Sims and Martin, 2014*; *Raggatt and Partridge, 2010*). Osteoclasts are giant multinucleated cells derived from myeloid macrophage lineage that function exclusively as bone resorbing cells. Osteoblasts are derived from bone marrow mesenchymal stem cells and differentiate into mature osteoblasts lining bone surface, and osteocytes embedded in bone. These cells are responsible for extracellular matrix formation and mineralization to form bone. Crosstalk between osteoclasts and osteoblasts/osteocytes coordinately couples their activities to ensure that osteoclast-generated resorption lacunae are filled with new bone produced by osteoblasts so as to maintain bone homeostasis during bone remodeling (*Sims and Martin, 2014*; *Raggatt and Partridge, 2010*). This provides an important

mechanism for adapting the skeleton to changing biomechanical influences and repairing bone damage throughout life.

The balance of this bone remodeling process, however, is disrupted in pathological conditions (*Goldring et al., 2013*; *Schett and Gravallese, 2012*; *Schett and Sieper, 2009*; *Walsh et al., 2009*). For example, excessive bone loss but limited bone formation often occur in postmenopausal osteoporosis, rheumatoid arthritis (RA), periodontitis, peri-prosthetic loosening, Paget's disease, and osteolytic bone tumors. On the other hand, excessive bone formation and/or defective bone erosion result in osteopetrosis or osteosclerosis. The mechanisms that regulate bone remodeling in physiological and pathological conditions are complex and far from well understood. In addition to diverse coupling factors, bone remodeling is also regulated by the molecular mechanisms that simultaneously control the differentiation and activities of the major cellular participants, osteoclasts, and osteoblasts/osteocytes. These mechanisms are underappreciated but important for understanding skeletal physiology and pathology as well as providing novel strategies to target both bone cell types, thereby effectively restoring normal remodeling for healthy skeleton. Recent joint association study of a genome-wide association study (GWAS) and a meta-analysis of bone mineral density (BMD) at different skeletal sites identified Differentially Expressed in FDCP 6 homolog (*DEF6*) as a novel loci associated with BMD (*Pei et al., 2019*). This new finding brought Def6 into the spotlight. Further studies are thus needed to elucidate the role of Def6 in bone remodeling and underlying mechanisms.

Def6, also known as IRF4-binding protein (IBP) or SWAP-70-like adaptor protein of T cells (SLAT), was originally identified as an activator of Rho GTPases with unique molecular structures and domains that have limited homology with other classical Rho-family guanine nucleotide exchange factors (GEFs) (*Gupta et al., 2003*; *Tanaka et al., 2003*). Def6 has a potential calcium binding EF-hand domain on the N terminus, followed by an immunoreceptor tyrosine-based activation motif-like sequence, a PI(3,4,5)P3-binding pleckstrin-homology (PH) domain, and a Dbl-homology (DH) domain exhibiting GEF activity on the C terminus (*Gupta et al., 2003*; *Mavrakis et al., 2004*; *Biswas et al., 2010*). This molecular structure is different from other typical GEFs, in which the GEF domain is on the N terminus of the PH domain. This molecular structure difference may contribute to the diverse biological functions of Def6 that distinguish it from other classical GEFs. Indeed, literature has shown that Def6 functions as a multifunctional protein that plays a variety of roles in immunology through diverse downstream effectors in addition to its GEF activity (*Biswas et al., 2010*; *Altman and Bécart, 2009*). Def6 is highly expressed in T cells and plays important roles in T cell proliferation, Th1/Th2 lineage differentiation and function. For instance, Def6 regulates IL-17 and IL-21 production in T cells via binding to and preventing the transcriptional activity of IRF4 (11, 13). Def6 associates with IP3 receptor 1 to regulate calcium signaling in T cells (*Fos et al., 2014*). Def6 is also expressed in myeloid cells and regulates innate immunity (*Chen et al., 2009*). Def6 deficient mice crossed with TCR transgenic DO11.10 mice develop RA-like joint disease with bone erosion (*Chen et al., 2008*). Both genetic mouse models and recent evidence obtained from patients (*Serwas et al., 2019*; *Sun et al., 2016*; *Yi et al., 2017*; *Manni et al., 2018*) show a critical role for Def6 in autoimmunity.

In addition to the immunoregulatory function of Def6, we have identified Def6 as a novel inhibitor of osteoclastogenesis in both physiological and inflammatory conditions (*Binder et al., 2017*). In RA patients, the serum TNF-α levels and disease activity are closely correlated with Def6 expression in osteoclast precursors. TNF-α downregulates Def6 expression in osteoclasts. Blocking the activity of TNF-α by TNF inhibitor treatment significantly increases Def6 expression levels, meanwhile decreasing osteoclast differentiation in RA osteoclast precursors, which supports an inhibitory role for Def6 in TNF-α induced osteoclastogenesis in humans. However, the role of Def6 in osteoblasts and bone formation has not been explored.

In the present study, we found that Def6 suppresses osteoblast differentiation and mineralization both in vitro and in vivo. Def6 knockout (KO) mice exhibit osteoporotic phenotype with enhanced osteoclast formation. Osteoblast differentiation and bone formation are elevated as well in Def6 KO mice, indicating a high bone turnover rate that leads to bone loss in Def6 KO mice. Thus, lack of Def6 leads towards unbalanced activities between osteoclastic resorption and osteoblast-mediated bone formation, disrupting normal bone remodeling. Recent studies from the osteoimmunology field highlight the critical role for the immune system in the regulation of bone homeostasis, mostly on bone resorbing osteoclasts (*Takayanagi, 2015*; *Goldring, 2013*; *Lorenzo and Choi, 2005*;

*Schett, 2009*; *Pacifici, 2013*). In this study, we found that Def6 suppresses osteoblast differentiation via endogenous type-I IFN response-mediated feedback inhibition, pointing to important immuno-regulation of osteoblasts and bone formation. These findings reveal that Def6 is a novel bone remodeling regulator in osteoimmunology that controls both osteoclast and osteoblast differentiation through immunoregulatory mechanisms to maintain bone remodeling.

## Results

### Def6 functions as an inhibitory regulator in osteoblast differentiation

Def6 is known as an inhibitor in osteoclastogenesis and bone resorption. To study whether Def6 regulates bone remodeling, we investigated the role of Def6 in osteoblast differentiation and bone formation in the present study. Def6 is expressed in calvarial osteoblast cells and is further induced during osteoblast differentiation and maturation (*Figure 1A,B*). The expression of Def6 in osteoblasts was also identified in vivo in mice as well as in osteoblast cell lines (*Figure 1—figure supplement 1*). This suggests that Def6 may have a function in osteoblasts. We first examined osteoblast differentiation in vitro using calvarial osteoblastic precursors obtained from the calvaria of the wild-type (WT) and *Def6$^{-/-}$* mice in the presence of osteogenic medium containing 10 mM β-glycerol phosphate and 100 μg/ml ascorbic acid. As shown in *Figure 1C*, Def6 deficiency significantly enhanced osteoblast differentiation and mineralization, evidenced by the marked increases in alkaline phosphate (ALP) activity (*Figure 1C*) and bone nodule formation determined by alizarin red staining (*Figure 1D*) in *Def6$^{-/-}$* cells. In parallel with the increased osteoblast differentiation, the expression of osteoblast marker genes, such as *Alpl* (encoding ALP), *Bglap* (encoding Osteocalcin) and *Bsp* (encoding Bone Sialoprotein), was significantly up-regulated in the *Def6$^{-/-}$* osteoblastic cultures compared to WT control cultures (*Figure 1E*). These results indicate that Def6 functions as a negative regulator in osteoblast differentiation.

### Def6 inhibits osteogenesis and bone formation in vivo

*Def6$^{-/-}$* mice exhibit osteoporotic phenotype shown in long bones (*Binder et al., 2017*) and vertebra (*Figure 2—figure supplement 1*). We then performed bone dynamic histomorphometric analysis by calcein double labeling to examine osteoblastic bone formation activity in these mice. As shown in *Figure 2A,B*, *Def6$^{-/-}$* mice displayed notably accelerated both mineral apposition rate (MAR) and bone formation rate (BFR/BS). *Def6$^{-/-}$* osteoblasts polarized to bone surface show clear cuboidal shape with thick osteoid formation beneath (*Figure 2C*). Furthermore, osteoblast parameters, such as osteoblast surface and numbers, were significantly up-regulated in *Def6$^{-/-}$* mice (*Figure 2D*). These results indicate that the lack of Def6 enhances osteogenesis and bone formation in mice. Therefore, both bone resorption and formation are enhanced by Def6 deficiency. The osteoporotic phenotype in *Def6$^{-/-}$* mice accordingly demonstrates a high bone turn-over rate that results in excessive bone loss. Hence, Def6 absence leads to unbalanced osteoclastic and osteoblastic activities, which disrupts normal bone remodeling and bone homeostasis.

### Def6 suppresses osteoblast differentiation via endogenous type-I IFN-mediated feedback inhibition

We next sought out to investigate the mechanisms by which Def6 suppresses osteoblast differentiation. To address this question, we first performed gene expression profiling using high-throughput sequencing of RNAs (RNAseq) with the WT control and *Def6$^{-/-}$* osteoblastic cells to identify genes regulated by Def6 during osteoblast differentiation. In this study, two biological RNAseq replicates were performed and analyzed for each condition. As shown in *Figure 3A*, Def6 deficiency significantly (p<0.05) up-regulated expression of 85 genes and down-regulated expression of 68 genes compared to WT cells during osteoblast differentiation (*Figure 3A*). Pathway analysis of the differentially expressed genes (DEGs) revealed that pathways most significantly activated in the WT and *Def6$^{-/-}$* osteoblasts were substantially different (*Figure 3B*). In alignment with the enhanced osteoblast differentiation and bone formation phenotype in *Def6$^{-/-}$* mice, the genes in pathways related to ossification and osteoblast were most significantly activated in *Def6$^{-/-}$* osteoblasts (*Figure 3B*). Surprisingly, in contrast to the *Def6$^{-/-}$* osteoblasts, the highly significant activation of genes in WT cells were involved in interferon signaling pathways, in particular type-I interferon signaling (*Figure 3B*).

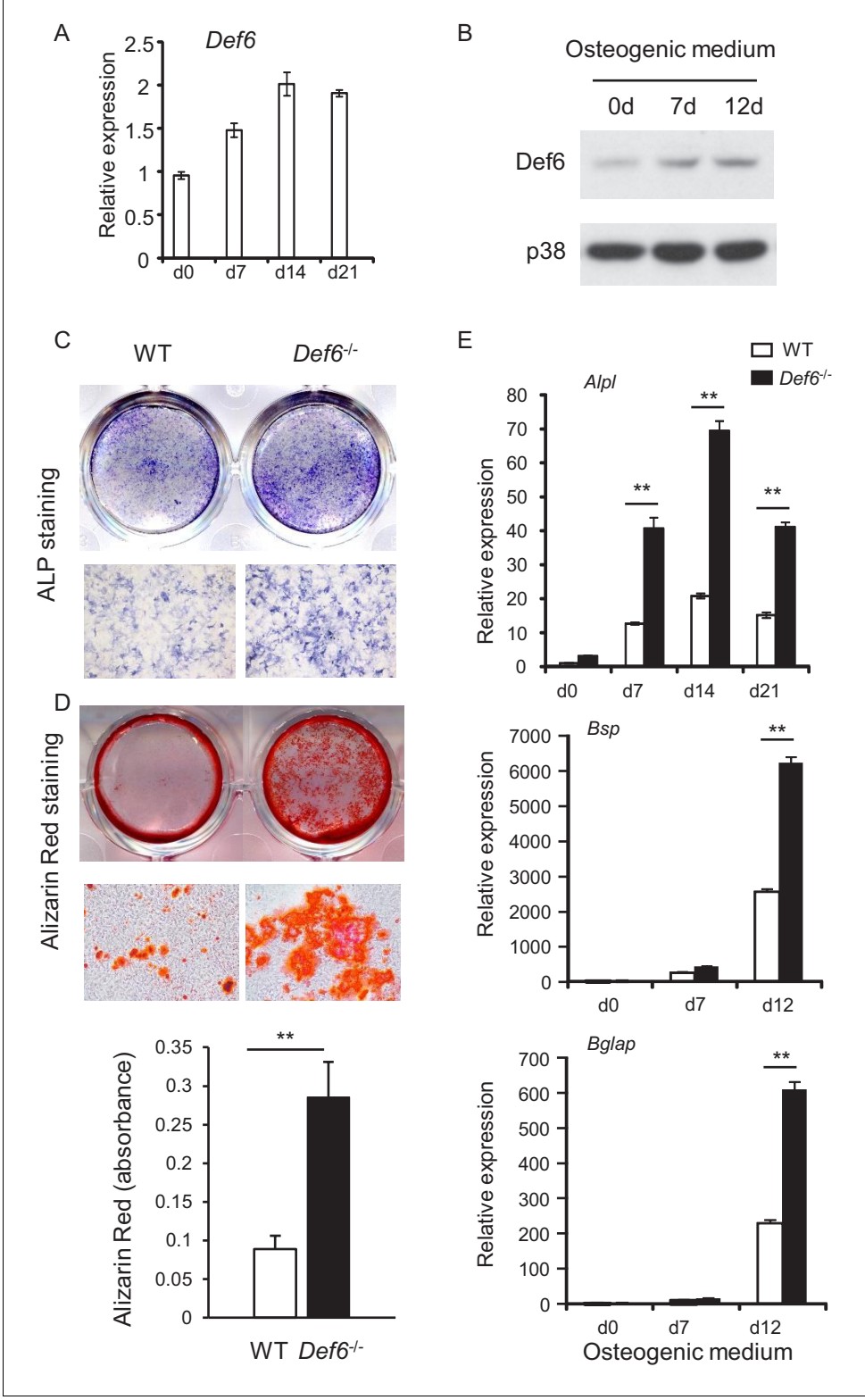

**Figure 1.** Def6 absence accelerates osteoblast differentiation. (**A, B**) Expression of Def6 during osteoblast differentiation using WT and *Def6*⁻/⁻ calvarial osteoblast cells at the indicated time points by quantitative real-time PCR (qPCR) analysis of mRNA expression (**A**) or immunoblot analysis of protein levels (**B**). (**C**) ALP staining and (**D**) Alizarin red staining (upper panel) and its quantification (lower panel) of WT and *Def6*⁻/⁻ calvarial osteoblast differentiation at day 15 in osteogenic medium. (**E**) qPCR analysis of mRNA expression of *Alpl*, *Bsp* and *Bglap*

*Figure 1 continued on next page*

*Figure 1 continued*

during WT and *Def6⁻/⁻* calvarial osteoblast differentiation process. Data are mean ± SEM. **p<0.01. n.s., not statistically significant.

The online version of this article includes the following source data and figure supplement(s) for figure 1:

**Source data 1.** Source data for Figure 1.

**Figure supplement 1.** Def6 expression in osteoblasts in vivo and in osteoblast cell lines.

**Figure supplement 2.** RNAseq–based expression heatmap of the marker genes of osteoclasts and macrophages (left panel) and volcano plot (right panel) of RNA-seq analysis of differentially expressed genes in WT and *Def6⁻/⁻* calvarial osteoblasts.

**Figure supplement 3.** CD45 selection does not affect Def6 role in osteoblast differentiation.

**Figure supplement 3—source data 1.** Source data for Figure 1—figure supplement 3.

**Figure supplement 4.** Bone marrow derived macrophages (BMMs) from WT and *Def6⁻/⁻* mice were co-cultured with CD45 negative WT calvarial osteoblasts using transwell inserts, in which BMMs were cultured in the upper compartment of the well and osteoblasts were on the bottom compartment of the well.

**Figure supplement 4—source data 1.** Source data for Figure 1—figure supplement 4.

**Figure supplement 5.** Immunoblot analysis of Def6 expression after TNFα (40 ng/ml) treatment for 24 hr on the WT and *Def6⁻/⁻* calvarial osteoblasts.

---

We further extracted the gene expression values of the most enriched type-I IFN response gene set from WT RNA-seq data and the osteoblast gene set from *Def6⁻/⁻* osteoblast RNA-seq data and displayed in the heatmaps in *Figure 3C*. Gene set enrichment analysis (GSEA) of the differentially expressed genes also revealed the most significantly enriched gene set to be type-I IFN response genes (p<0.0001 and FDR < 0.001) in WT osteoblasts compared to *Def6⁻/⁻* cells. Similarly, osteoblast genes were enriched (p<0.0001 and FDR < 0.001) in *Def6⁻/⁻* cells (*Figure 3D*). We further confirmed the elevated expression of osteoblast marker genes, such as *Alpl*, *Bglap*, and *Bsp*, in *Def6⁻/⁻* cells (*Figure 1E*), and type-I IFN response genes, such as *Mx1*, *Ifit1*, *Ifit2*, *Cxcl10*, *Stat1* and *Eif2ak2* (encoding Protein kinase RNA-activated (PKR)), in WT osteoblasts (*Figure 3E*). These results suggest that an interferon response as demonstrated by interferon stimulated gene (ISG) expression occurs during osteoblastogenesis, and this IFN response and ISG expression are positively regulated by Def6.

Since type-I IFN signaling components are widely expressed, and most cells are competent to type-I IFN response (*Ivashkiv and Donlin, 2014*), we investigated whether the ISG gene expression was induced by endogenous IFNβ secreted by osteoblasts. The cellular type-I IFN activity, especially induced by low or undetectable level of type-I IFN, is usually reflected by ISG expression (*Ivashkiv and Donlin, 2014*). We blocked IFNβ activity using an IFN-β neutralizing antibody and found that the ISG gene expression was significantly decreased in both WT and *Def6⁻/⁻* cell cultures (*Figure 4A*). This data supports the idea that the ISG gene expression in osteoblasts is induced, at least partially, by endogenous IFNβ. Previous literature show that a small to undetectable magnitude of type-I IFN can have high potency of biological effects, such as the conditions that are involved in osteoclast inhibition and 'IFN signature' of ISG expression in systemic lupus erythematosus (SLE) (*Binder et al., 2017*; *Ivashkiv and Donlin, 2014*; *Takayanagi et al., 2002*; *Inoue et al., 2018*). Indeed, when the IFNβ activity was blocked, the expression of osteoblast differentiation marker genes, such as *Bsp* and *Bglap*, was significantly elevated in WT cultures (*Figure 4B*). This enhancement in WT cells is much higher than that in *Def6⁻/⁻* cell cultures (*Figure 4B*). For example, the *Bsp* gene expression in WT cells was enhanced to a similar level to that in *Def6⁻/⁻* cells by IFNβ blocking antibody. The enhancement of *Bglap* gene expression by IFNβ blocking antibody was 5.4 folds in WT cells vs 1.5 folds in *Def6⁻/⁻* cell cultures. These differences corroborate a lower IFNβ activity feature in *Def6⁻/⁻* cell cultures. In alignment with the osteoblast marker gene expression, blocking IFNβ activity enhanced osteoblast differentiation and enabled the differentiation in WT cells to reach to a similar level to that in *Def6⁻/⁻* cell cultures (*Figure 4C*). Therefore, the endogenous IFNβ activity plays an inhibitory feedback role in osteoblast differentiation. Taken together that the genome-wide data shows that the most differentially activated genes by Def6 are type-I IFN response genes (*Figure 3*), these results collectively indicate that the endogenous IFNβ-mediated response is a prominent mechanism by which Def6 inhibits osteoblast differentiation.

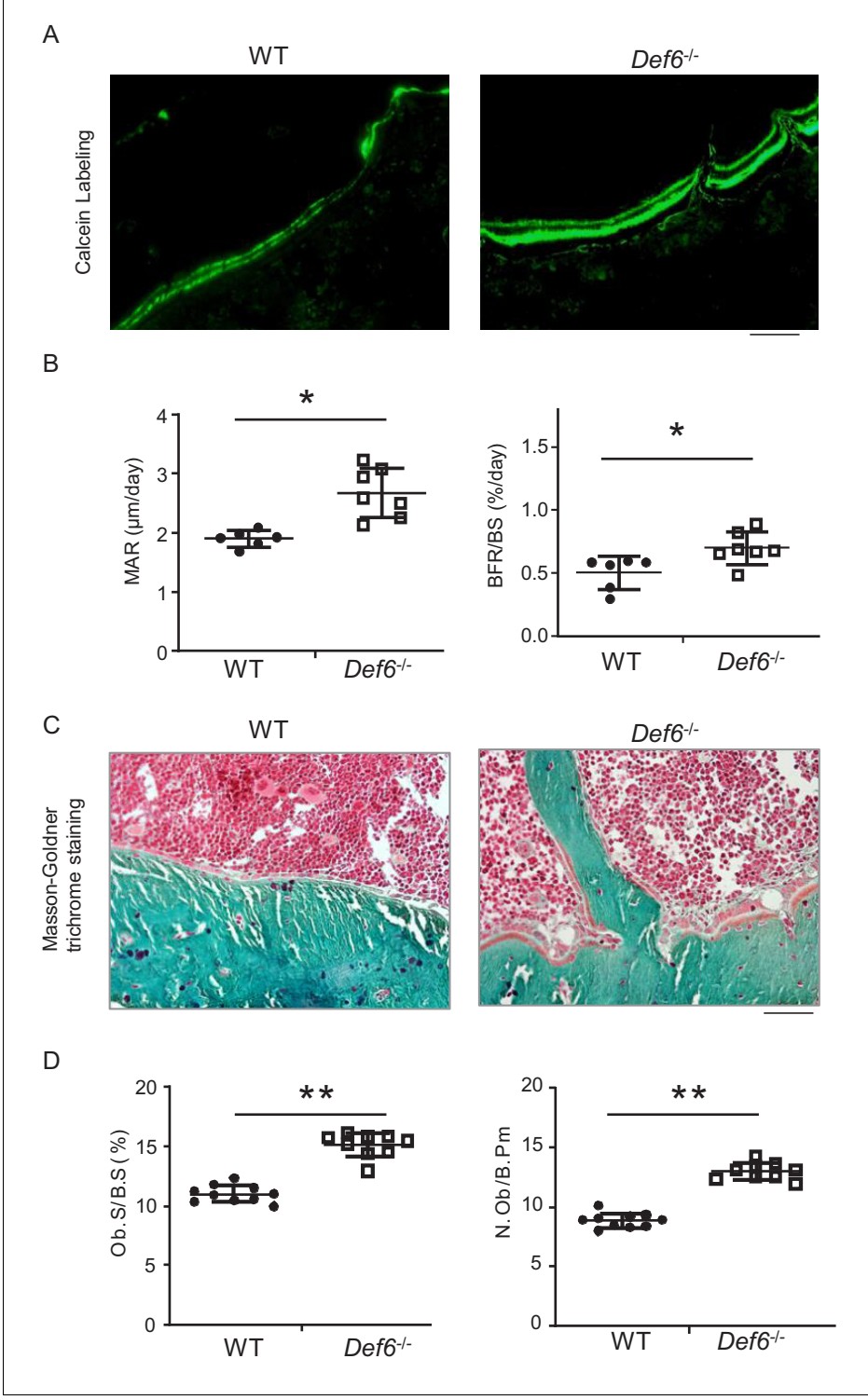

**Figure 2.** Def6 deficiency enhances osteogenesis in vivo. (**A**) Images of calcein double labeling of the tibiae of WT and *Def6⁻/⁻* littermate mice. (**B**) Bone morphometric analysis of mineral apposition rate (MAR) and bone formation rate per bone surface (BFR/BS) after calcein double labeling of the tibiae of WT and *Def6⁻/⁻* littermate male mice. n = 6. (**C**) Images of Masson-Goldner staining of tibiae of WT and *Def6⁻/⁻* littermate mice. The bones show green, osteoid matrix appears dark orange on the surface of the bone beneath the osteoblasts, osteoblasts (Ob) are stained orange lining on the bone surface, and bone marrow cells appear red in the photograph. (**D**) Bone morphometric analysis of osteoblast surface per bone surface (Ob.S/BS) and osteoblast number per bone

*Figure 2 continued on next page*

*Figure 2 continued*

perimeter (N.Ob/B.Pm) of the tibiae of WT and *Def6⁻/⁻* littermate male mice. n = 8. Scale bars = 50 μm. Data are mean ± SEM. *p<0.05, **p<0.01. n.s., not statistically significant.

The online version of this article includes the following source data and figure supplement(s) for figure 2:

**Source data 1.** Source data for Figure 2.
**Figure supplement 1.** Def6 exhibits osteoporosis in lumbar bones.
**Figure supplement 1—source data 1.** Source data for Figure 2—figure supplement 1.

WT and *Def6⁻/⁻* osteoblasts cells express comparable levels of interferon-α/β receptor (**Figure 5A** and **Figure 5—figure supplement 1**). We next tested the cellular response of osteoblasts to type-I IFN by adding IFNβ to the cell cultures. As shown in **Figure 5B,C**, IFNβ stimulation induced phosphorylation of Stat1 and Stat3 in WT osteoblastic cells. In contrast, Def6 deficiency dampened the activation levels of Stat1 and Stat3 in *Def6⁻/⁻* cells (**Figure 5B,C**). Stat1 is a type-I IFN inducible gene. The expression of Stat1 was reduced by Def6 absence (**Figure 5D**). These results indicate that *Def6⁻/⁻* osteoblastic cells exhibit attenuated IFN responsiveness, which was further corroborated by the decreased ISG expression, such as *Mx1*, *Ifit2 and Irf7*, in *Def6⁻/⁻* osteoblastic cell cultures (**Figure 5E**). To test whether the attenuated type-I IFN response contributes to cell differentiation, we treated osteoblasts with IFNβ, and found that the differentiation of WT cells was drastically inhibited by IFNβ, but the inhibitory effect of IFNβ on osteoblast differentiation in *Def6⁻/⁻* cells was much weaker (**Figure 5F**). These results indicate that Def6 deficiency attenuates cellular response to IFN-induced osteoblastic inhibition.

Collectively, Def6 functions as an inhibitor that suppresses osteogenesis via endogenous type-I IFN-mediated feedback inhibition in physiological conditions (**Figure 5G**). Our results identify Def6 as an upstream regulator of type-I IFN response in both osteoclast (**Binder et al., 2017**) and osteoblast differentiation (present study). Thus, Def6-IFN-I axis is a common and key mechanism involved in both osteoclast and osteoblast differentiation to maintain physiological bone remodeling. Lack of Def6 leads to suppressed type-I IFN response, which in turn promotes the differentiation of osteoclasts and osteoblasts, resulting in a high turn-over osteoporosis (**Figure 5G**).

## Def6-PKR/IFNβ axis plays an important role in feedback inhibition of osteoblastogenesis

We next examined the expression levels of IFNα and IFNβ in mice. The serum IFNβ level is significantly lower in *Def6⁻/⁻* mice than WT mice (**Figure 6A**). Importantly, Def6 deficiency significantly decreased IFNβ level in osteoblast cells (**Figure 6—figure supplement 1**) as well as in osteoblast culture medium (**Figure 6B**). IFNα level is nearly undetectable in the serum from WT or *Def6⁻/⁻* mice, and undetectable in osteoblast culture medium (**Figure 6—figure supplement 2** and data not shown). These results demonstrate that osteoblasts produce IFNβ, and Def6 deficiency downregulates the endogenous IFNβ level secreted by osteoblasts. We have shown that exogenous IFNβ inhibits osteoblastogenesis (**Figure 5F**). Blocking endogenous IFNβ activity by IFNβ blocking antibody significantly enhances osteoblast differentiation (**Figure 4**). We further examined the effect of IFNβ on osteoblast differentiation using *Ifnar1⁻/⁻* cells, which do not activate downstream type-I IFN signaling. As shown in **Figure 6C–E**, deficiency of type-I IFN receptor significantly enhanced osteoblastogenesis, evidenced by the increases in ALP activity (**Figure 6C**), bone nodule formation (**Figure 6D**) and osteoblast marker gene expression, such as *Alpl* and *Bglap* (**Figure 6E**). As expected, the expression of ISG genes, such as *Mx1*, *Ifit1* and *Irf7*, was markedly induced in the WT osteoblast cultures, but abolished in *Ifnar1⁻/⁻* osteoblasts (**Figure 6E**). These results collectively corroborate the presence of endogenous IFNβ and IFNβ response during osteoblastogenesis, which provides another example that underscores the significant biological role of low level of IFNβ. Our data also demonstrates that endogenous IFNβ plays an inhibitory role in osteoblast differentiation.

As Def6 deficiency significantly enhances osteoblastogenesis via attenuating IFNβ level and its osteoblastic inhibition, we then set off to investigate how Def6 regulates IFNβ expression and its downstream IFNβ response. We found that loss of Def6 decreased PKR expression in osteoblasts (**Figures 3E** and **6F**). PKR (Protein kinase RNA-activated, gene name: *Eif2ak2*) is an ISG gene and also functions as an activator of IFN-β expression (**Haller et al., 2006**; **Munir and Berg, 2013**;

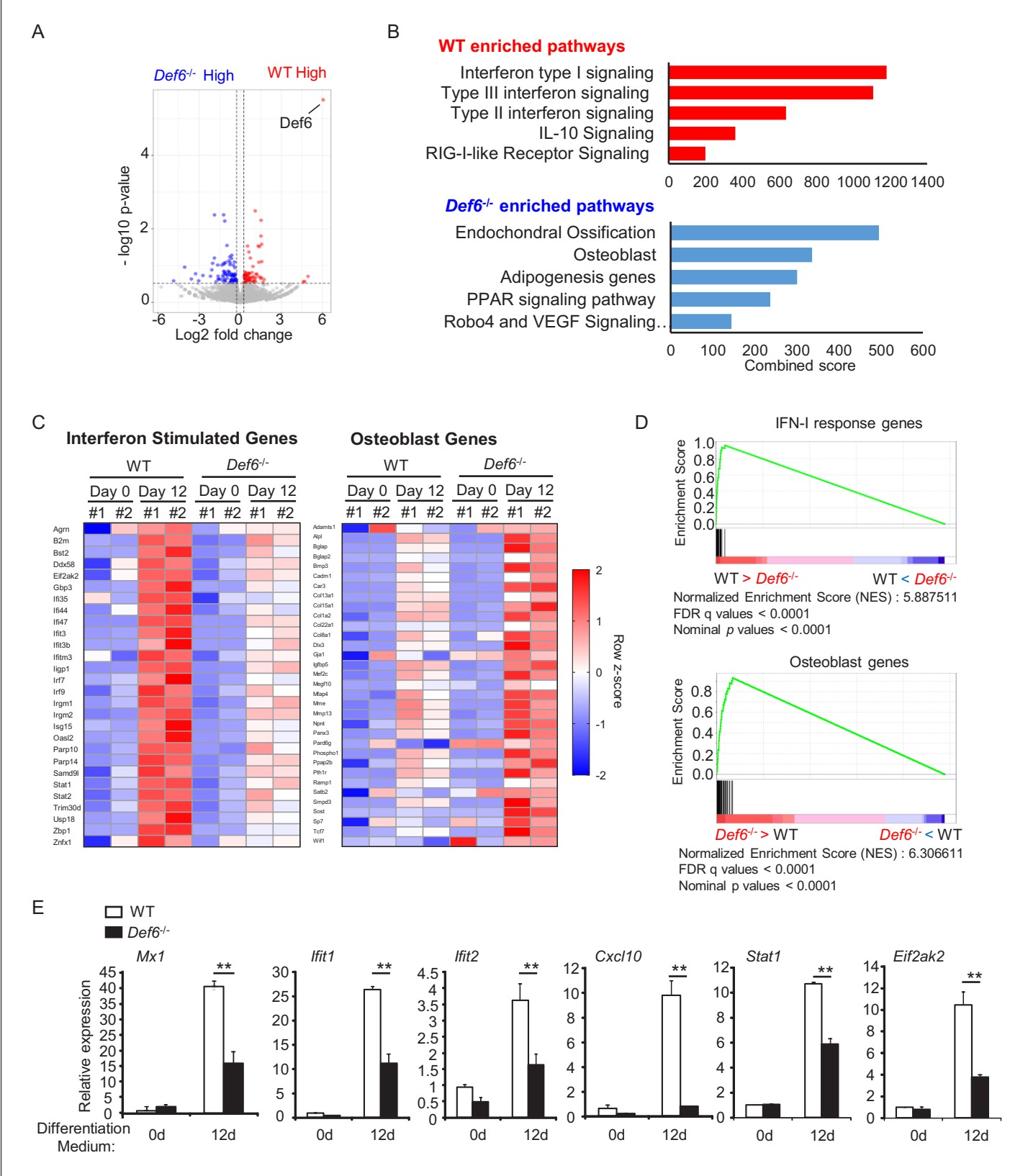

**Figure 3.** Def6 deficiency decreases the expression of interferon stimulated genes (ISGs), which are induced during osteoblast differentiation. (**A**) Volcano plot of RNA-seq analysis of differentially expressed genes using the mRNAs isolated from the WT and *Def6*^-/-^ calvarial osteoblast cells stimulated with osteogenic medium for 12 days. Blue dots show genes more highly expressed in *Def6*^-/-^ osteoblasts than WT osteoblasts with

*Figure 3 continued on next page*

*Figure 3 continued*

significant (p<0.05) and greater than 1.2-fold changes. Red dots show genes more highly expressed in WT osteoblasts than *Def6⁻/⁻* osteoblasts with significant (p<0.05) and greater than 1.2-fold changes. (**B**) Pathway analysis of significantly differentially expressed genes. Upper panel shows enriched pathways in WT and lower panel shows enriched pathways in *Def6⁻/⁻* osteoblasts. (**C**) RNAseq–based expression heatmaps of the interferon stimulated genes (left panel) and osteoblast genes (right panel) regulated by Def6 deficiency based on the differentially expressed genes from (**A**). Row z-scores of CPMs of genes were shown in the heatmap. #1, replicate 1. #2, replicate 2. (**D**) Gene set enrichment analysis (GSEA) of differentially expressed genes from WT and *Def6⁻/⁻* calvarial osteoblast cells stimulated with osteogenic medium for 12 days ranked by NES scores. Type-I IFN response genes and osteoblast genes are mostly enriched in WT and *Def6⁻/⁻* osteoblast cells, respectively (p<0.0001 and FDR < 0.0001). The enrichment plots are shown. NES, normalized enrichment score. (**E**) qPCR analysis of type-I IFN response gene expression during osteoblast cell differentiation. Data are mean ± SEM. **p<0.01. n.s., not statistically significant.

The online version of this article includes the following source data for figure 3:

**Source data 1.** Source data for Figure 3.

*Meurs et al., 1990*; *McAllister et al., 2012*; *Taghavi and Samuel, 2012*). We knocked down PKR expression in ST2 cells (*Figure 6G*), an osteoblast cell line well-established for osteoblast differentiation and mechanistic studies. Similarly as calvarial osteoblasts, the expression of *Ifnb* and ISG genes was highly induced during osteoblast differentiation in ST2 cells (*Figure 6H* white bars). PKR deficiency resulted in a drastic decrease in the expression of *Ifnb* and ISG genes, such as *Mx1, Ifit1, Stat1 and Irf7* (*Figure 6H* black bars), indicating that PKR is a key activator for IFN-β and downstream ISG gene induction in osteoblasts. In contrast, PKR deficiency significantly promoted osteoblast differentiation (*Figure 6I*) and the expression of osteoblastic genes, such as *Runx2, Alpl, Bsp and Bglap* (*Figure 6J*). Taken together, these findings indicate that PKR controls endogenous IFN-β production and response in osteoblasts, thereby contributing to the feedback inhibitory effects mediated by endogenous IFN-β. Def6 deficiency significantly downregulates PKR expression level (*Figures 3E* and *6F*). Furthermore, overexpression of PKR in Def6 deficient osteoblasts enhances *Ifnb* and ISG expression but suppresses the expression of osteoblastic marker genes, such as *Alpl and Bsp* (*Figure 6K–L*). Thus, Def6 deficiency suppresses IFN-β expression, at least partially via downregulation of PKR expression. Therefore, our results unveiled a new pathway mediated by Def6-PKR/IFN-β axis that plays an important role in feedback inhibition of osteoblast differentiation.

## Discussion

The adult skeleton undergoes constant and dynamic remodeling process throughout life to maintain bone homeostasis, which is achieved by the tight control of coupling between osteoclast-mediated bone resorption and osteoblast-mediated bone formation (*Sims and Martin, 2014*; *Raggatt and Partridge, 2010*). The studies of cell differentiation and function for each cell type are extensive. However, the mechanisms that simultaneously regulate both cell types during bone remodeling are less appreciated. Understanding of these mechanisms can help identify novel therapeutic targets for regulating the activity of both cell types involved in bone remodeling, and thereby facilitating and enhancing treatment efficacy. Our previous work identified Def6 as an important inhibitory regulator for osteoclastogenesis and bone resorption. In this study, we found that Def6 is an inhibitor that suppresses osteoblastic differentiation and osteogenesis in vivo. Therefore, Def6 is a common regulator that negatively regulates both osteoclastogenesis and osteoblastogenesis simultaneously in bone remodeling. Def6 deficiency enhances both osteoclastic bone resorption and osteoblastic bone formation. However, the regulatory potency by Def6 in the two cell types is unbalanced, leading to a high turnover osteoporotic phenotype in *Def6⁻/⁻* mice. Hence, Def6 is an important regulator that maintains physiological bone homeostasis. Environmental cues or pathological conditions that affect Def6 expression and activity may deregulate bone remodeling and impact bone mass.

In this study, we utilized a well-established and widely used in vitro osteoblast culture system, in which the osteoblastic cells were isolated from the calvaria of newborn mice. Since Def6 plays an inhibitory role in macrophage differentiation into osteoclasts, we validated whether the calvarial osteoblast culture system is solid to investigate the role of Def6 in osteoblastogenesis without significant macrophage lineage effects. Using RNAseq, a sensitive approach, we can only detect very low expression levels (below 1–2 counts per million reads) of marker genes for either macrophages or osteoclasts in this culture system. Furthermore, these genes belong to non-significant genes and

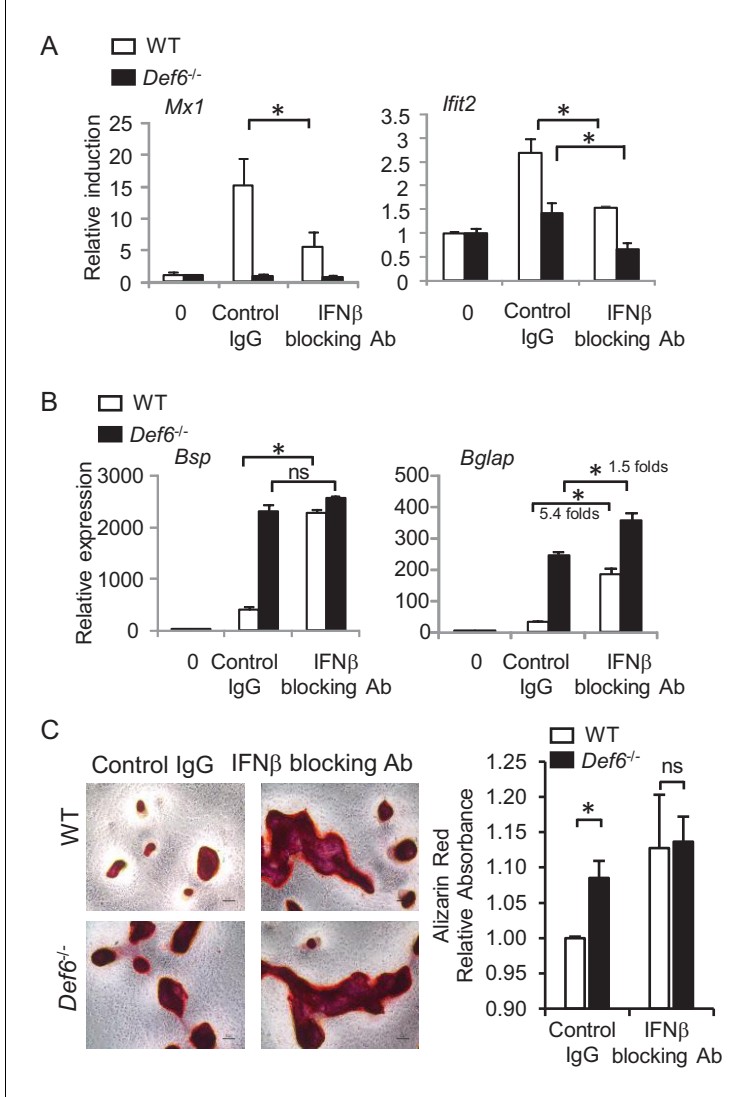

**Figure 4.** Def6 suppresses osteogenesis via endogenous IFNβ-mediated autocrine feedback inhibition. qPCR analysis of type-I IFN response gene expression (**A**) and osteoblast marker gene expression (**B**) during osteoblast cell differentiation in the absence or presence of IFNβ blocking antibody (10 U/ml). Control IgG, 10 U/ml. (**C**) Alizarin red staining (left panel) and its quantification (right panel) of WT and *Def6^-/-* calvarial osteoblast differentiation at day 16 in the osteogenic medium in the absence or presence of IFNβ blocking antibody (10 U/ml). Control IgG, 10 U/ml. Data are mean ± SEM. *p<0.05. n.s., not statistically significant.

The online version of this article includes the following source data for figure 4:

**Source data 1.** Source data for Figure 4.

there are no differences in their expression between WT and Def6 deficient osteoblast cultures (*Figure 1—figure supplement 2*). To exclude any potential effect from hematopoietic cells, we removed CD45 positive cells, which label all hematopoietic cells, and selected CD45 negative calvarial osteoblasts to compare the osteoblast differentiation. Similarly as the results in non-select cultures shown in *Figures 1* and *3*, Def6 deficiency significantly enhanced osteoblastogenesis and osteoblast marker gene expression, but reduced ISG gene expression without CD45 positive cells in cultures (*Figure 1—figure supplement 3*). Moreover, we co-cultured CD45 negative calvarial osteoblasts with bone marrow derived macrophages (BMMs) isolated from WT or Def6 deficient mice. We found that Def6 deficiency in BMMs does not affect osteoblast differentiation (*Figure 1—figure supplement 4*). Therefore, these results collectively validated this well-established osteoblast culture system, which is solid to test Def6 function in osteoblast differentiation.

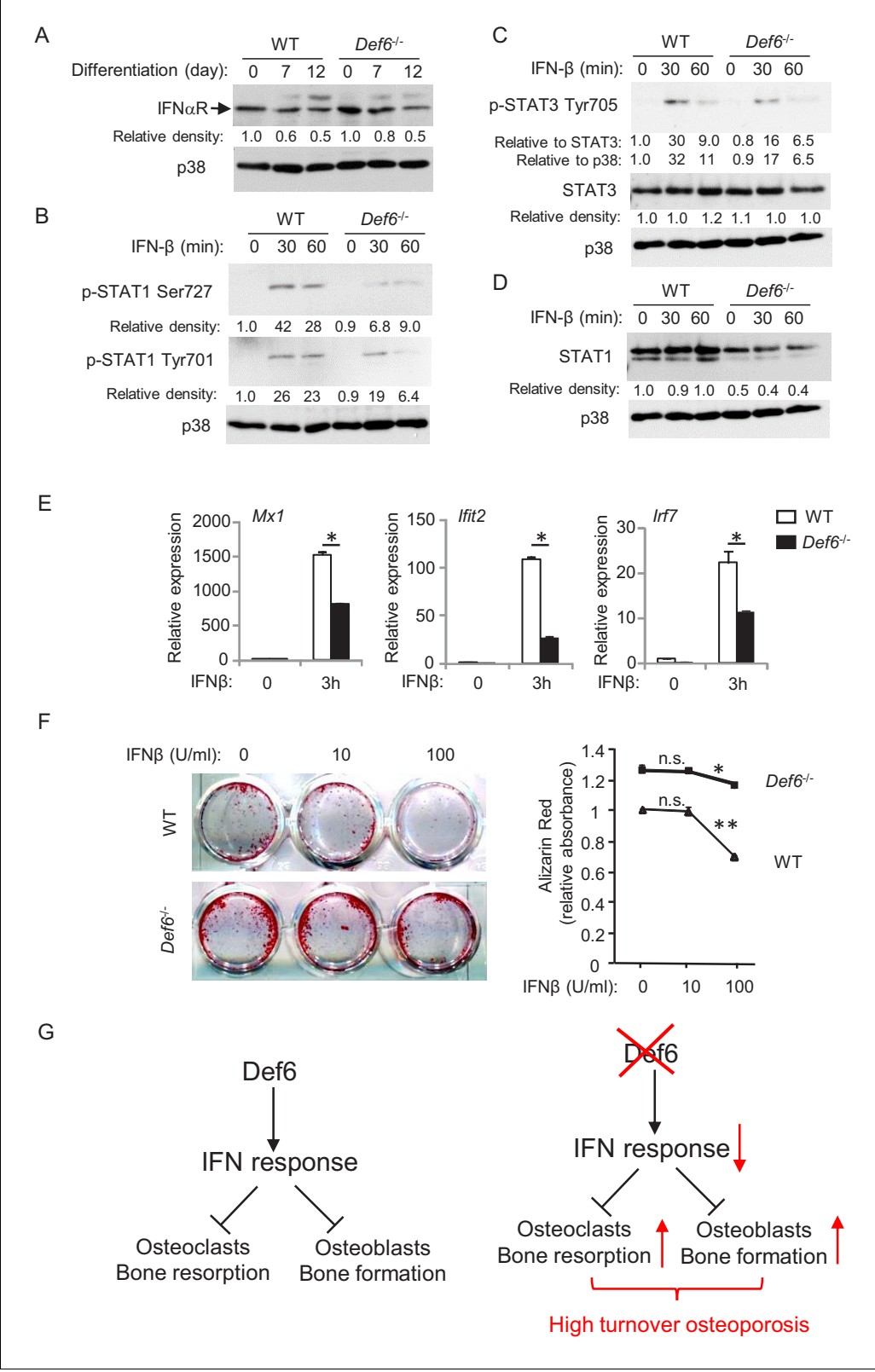

**Figure 5.** Def6 regulates cellular response of osteoblast cells to IFNβ. (A–D) Immunoblot analysis of type-I IFN receptor expression during osteoblast differentiation (A), the induction of p-STAT1 and 3 (B, C) and expression of STAT1 (D) in WT and *Def6*⁻/⁻ osteoblast cells stimulated with mouse recombinant IFN-β (10 U/ml) at the indicated time points. p38 was used as a loading control. The relative density of each band to its corresponding loading
*Figure 5 continued on next page*

*Figure 5 continued*

control p38 band was calculated by Image J software, and then was normalized to the WT controls at time 0 (the 1st lanes). The relative density for p-STAT3 to total STAT3 was also calculated. (E) qPCR analysis of type-I IFN response gene expression after indicated time points of treatment with IFN-β (10 U/ml) in WT and Def6$^{-/-}$ osteoblast cells. (F) Osteoblast differentiation is induced by the osteogenic medium in the absence or presence of mouse recombinant IFN-β in the WT and *Def6*$^{-/-}$ osteoblast cells. Alizarin red staining at day 16 (left panel) and its quantification (right panel) were performed. Data are mean ± SEM. *p<0.05, **p<0.01. n.s., not statistically significant. (G) A model showing that the Def6-IFN axis regulates both osteoclast-mediated bone resorption (*Binder et al., 2017*) and osteoblast-mediated bone formation (current study) in bone homeostasis. Def6 deletion enhances both bone resorption (*Binder et al., 2017*) and formation (current study) via attenuated type-I IFN-mediated feedback inhibition of the differentiation of both cell types, leading to a high turn-over osteoporotic phenotype in *Def6*$^{-/-}$ mice.

The online version of this article includes the following source data and figure supplement(s) for figure 5:

**Source data 1.** Source data for Figure 5.
**Figure supplement 1.** qPCR analysis of mRNA expression of *Ifnar1* and *Ifnar2* during osteoblast differentiation.
**Figure supplement 1—source data 1.** Source data for Figure 5—figure supplement 1.

---

Accumulating evidence shows that Def6 acts as a multifunctional protein through its GEF activity or binding to other regulators, such as IRF4 and IP3 receptor 1, to regulate the activities of immune cells including T cells and myeloid cells. A large body of work has also established the immunoregulatory function of Def6 in innate and adaptive immunity as well as in autoimmunity. The studies (*Binder et al., 2017*, and current study) from our group revealed an important novel role for Def6 in maintaining bone homeostasis, in which Def6 functions as an inhibitor that suppresses the differentiation of both osteoclasts and osteoblasts. Interestingly, we found that Def6 executes this inhibitory function via endogenous type-I IFN-mediated feedback inhibition of bone cell differentiation, the same downstream mechanisms by which Def6 suppresses differentiation of both osteoclasts and osteoblasts. In the osteoclast differentiation process, it is well documented that RANKL or TNFα treatment induces type-I IFN expression in macrophages/osteoclast precursors. Although the magnitude of type-I IFN induction by RANKL or TNFα is small (low picomolar concentrations) when compared with other stimuli such as toll-like receptor (TLR) stimulation or viral infection, the endogenous autocrine type-I IFNs show significant feedback inhibitory effects on osteoclast differentiation (*Binder et al., 2017*; *Takayanagi et al., 2002*; *Inoue et al., 2018*). Our group and others have shown that activation of type-I IFN signaling by exogenous IFNβ suppresses osteoblast differentiation, matrix formation and mineralization (*Woeckel et al., 2012a*; *Woeckel et al., 2012b*; *Kim et al., 2003*). However, the effects of endogenous or basal IFN activity in osteoblastic cells on their differentiation have recently drawn attention with variable reports from literature (*Takayanagi et al., 2002*; *Kota et al., 2018*). We and other groups (*Kota et al., 2018*) recently detected endogenous ISG expression in osteoblastic culture system in the absence of virus, pathogens or exogenous IFNβ. Interestingly, ISG gene expression is induced along osteoblast differentiation, but is reversely correlated with the osteogenic potential between different cultures, for example, *Def6*$^{-/-}$ osteoblastic cells express lower ISGs but have higher osteogenic potential than WT cells. The puzzle is what is the inducer of the ISG expression in osteoblastic cells. When we added IFNβ blocking antibody to the cultures to block potential IFNβ, we found that the ISG expression was largely decreased, indicating that the osteoblastic cells secrete IFNβ, which subsequently induces ISG expression. Indeed, our ELISA results verified autocrine IFNβ, but not IFNα, in the osteoblast cultures. Despite that the amount of autocrine IFNβ is low (below 5 pg/ml), its biological function in ISG induction and osteoblastic inhibition is clearly present. This is not surprising for homeostatic conditions (*Gough et al., 2012*) or certain autoimmune diseases, such as SLE, in which an 'IFN signature' of ISG expression is observed although IFNβ may not be detectable (*Ivashkiv and Donlin, 2014*). Studies reveal the existence of constitutive type-I IFN in healthy mice that are maintained in specific pathogen-free environments. Constitutive type-I IFNs are usually produced at vanishingly low amounts in the absence of infection and yet have a variety of biological effects, such as maintenance of the hematopoietic stem cell (HSC) niche, regulation of immune cell function and osteoclastogenesis (*Gough et al., 2012*). These highlight the important biological roles for endogenous IFN in cells/tissues. Since the expression levels of endogenous type-I IFN are usually low, close to the threshold

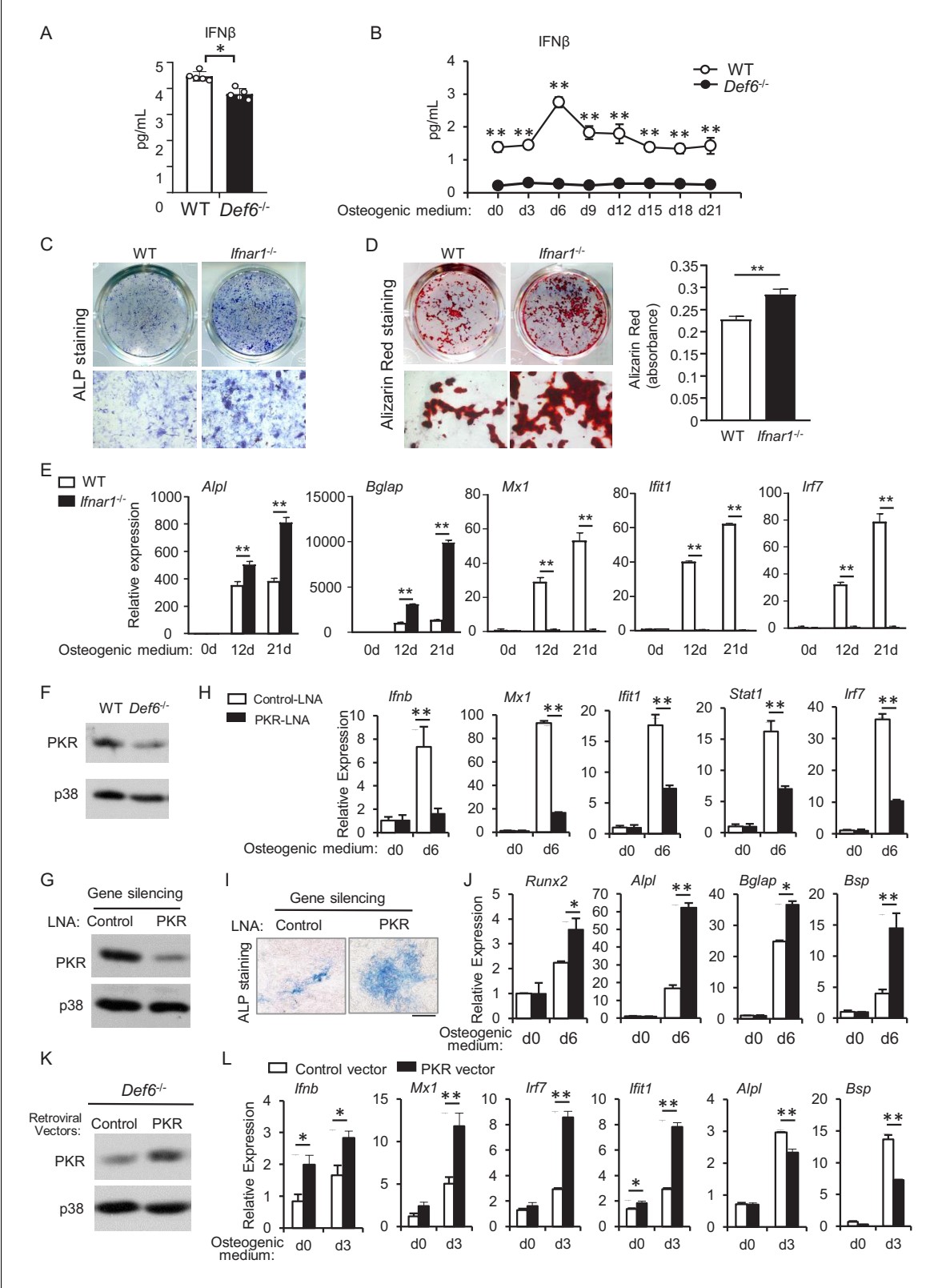

**Figure 6.** Def6 deficiency inhibits endogenous expression of IFN-β and ISG genes via downregulation of PKR in osteoblasts. (A–B) ELISA analysis of IFN-β level in mouse serum (A) and in CD45 negative calvarial osteoblast cell cultures (B) from WT and *Def6*^-/-^ mice (n = 5). (C) ALP staining at day 7 and (D) Alizarin red staining at day 21 (left panel) and its quantification (right panel) of WT and *Ifnar1*^-/-^ (IFNαβR KO) calvarial osteoblast differentiation in osteogenic medium. (E) qPCR analysis of mRNA relative expression of osteoblast marker genes and ISGs during WT and *Ifnar1*^-/-^ (IFNαβR KO)

*Figure 6 continued on next page*

*Figure 6 continued*

calvarial osteoblast differentiation process. (F) Immunoblot analysis of PKR expression in WT and *Def6*⁻ᐟ⁻ calvarial osteoblasts. p38 was used as a loading control. (G) Immunoblot analysis of PKR expression after knockdown of PKR by PKR-LNA induced gene silencing in ST2 cells. p38 was used as a loading control. (H–J) qPCR analysis of mRNA expression of *Ifnb* and ISGs (H), ALP staining, scale bar: 500 μm (I) and qPCR analysis of mRNA expression of osteoblastic genes (J) in ST2 cells with or without PKR knockdown. (K) Immunoblot analysis of PKR expression after overexpression of PKR by retroviral transduction for 24 hr. p38 was used as a loading control. (L) qPCR analysis of mRNA expression of *Ifnb*, ISGs and osteoblast markers. Data are mean ± SD. *p<0.05. **p<0.01. n.s., not statistically significant.

The online version of this article includes the following source data and figure supplement(s) for figure 6:

**Source data 1.** Source data for Figure 6.
**Figure supplement 1.** qPCR analysis of mRNA expression of *Ifnb* during osteoblast differentiation.
**Figure supplement 1—source data 1.** Source data for Figure 6—figure supplement 1.
**Figure supplement 2.** ELISA analysis of IFNα level in the serum of 8 week old WT and *Def6*⁻ᐟ⁻ mice.
**Figure supplement 2—source data 1.** Source data for Figure 6—figure supplement 2.

of detection, the IFN activity is also commonly indicated and reflected by the ISG expression in these conditions. Our results thus unveil an important novel immunoregulatory function of Def6 in bone cells, in which Def6 acts as an upstream regulator of IFNβ and ISG gene expression and endogenous type-I IFN response to regulate bone cell differentiation. These findings highlight the biological importance of the close interaction of immune system and bone health.

Induction of *Ifnb1* expression involves, in most cases, the activation of AP-1, NF-κB and various interferon regulatory factors (IRFs), depending on the cell types and context of the stimulation (*Pitha and Kunzi, 2007*). In response to strong IFN inducers, such as virus or TLR stimulation, *Ifnb1* induction often occurs rapidly and strongly within minutes to a few hours and is dependent on IRF3/7 (*Pitha and Kunzi, 2007*). In contrast, *Ifnb1* induction in osteoclast or osteoblast cultures occurs slowly and mildly. TNFα–induced IFNβ expression in monocytes/macrophages is mainly dependent on IRF1 (*Yarilina et al., 2008*) and Def6 (*Binder et al., 2017*). RANKL induces IFNβ expression via c-Fos and Def6 during osteoclastogenesis (*Binder et al., 2017*; *Takayanagi et al., 2002*). This RANKL-induced autocrine IFNβ expression is negatively controlled by miR182-PKR axis (*Inoue et al., 2018*). The attention on the endogenous autocrine IFNβ loop is just emerging. How IFNβ is induced in osteoblastic cells was unknown. We, for the first time, identified Def6 as an upstream regulator that modulates endogenous IFNβ expression and activity, significantly via PKR, during osteoblastogenesis. PKR was originally identified as an IFNβ inducible gene in response to viral infection, and plays an important role in type-I IFN production (*Haller et al., 2006*; *Munir and Berg, 2013*; *Meurs et al., 1990*; *McAllister et al., 2012*; *Taghavi and Samuel, 2012*). Various downstream pathways, such as IFNβ and eIF2a-mediated translational inhibition, can be regulated by PKR-dependent on cell types and context (*Yang et al., 2010*; *Opitz et al., 2009*; *Lourenco et al., 2013*; *Hsu et al., 2004*). In osteoblasts, we did not observe phosphorylation of eIF2α (data not shown). Instead, IFNβ was identified as a significant downstream factor mediated by PKR in osteoblasts, corroborated by our data that PKR deficiency almost abrogates the expression of IFNβ and downstream ISG genes. On the other hand, PKR, as an IFNβ inducible gene, could also be regulated by the autocrine IFNβ in the osteoblast cultures, leading to a regulatory circuit between PKR and IFNβ downstream of Def6 that coordinately mediate osteoblastogenesis. In addition, Stat1 is an ISG gene and previous work demonstrates its inhibitory function in osteoblast differentiation by binding to and preventing the activity of Runx2 (*Kim et al., 2003*). A recent study shows that Gbp1, an ISG gene, inhibits osteogenic differentiation (*Bai et al., 2018*). It seems that the ISG genes play important roles in osteoblastic inhibition, which is distinct from their originally identified anti-viral function. Def6 deficiency significantly decreases a group of ISG gene expression, including Stat1 and Gbp3, which belongs to Gbp family. Thus, Def6 may also function through these ISGs to suppress osteogenesis. It would be of great interest for future studies to fully dissect the underlying molecular mechanisms by which PKR and other ISGs regulate osteoblast differentiation and bone formation.

Recent human genetic evidence, together with results obtained from animal models and patients, reveal the important regulatory role for Def6 in immunity and autoimmune diseases, such as SLE and RA. Def6 deficient female mice on a mixed 129/BL6 background develop a lupus-like syndrome (*Biswas et al., 2010*; *Fanzo et al., 2006*). Def6 is a new systemic lupus erythematosus (SLE) risk variant (*Sun et al., 2016*) and recent evidence obtained from patients show a critical role for Def6 in

autoimmunity (*Serwas et al., 2019*). In addition to the role for Def6 in immune diseases, recent GWAS study identifies *DEF6* as a novel loci associated with BMD (*Pei et al., 2019*). Along this line, our results show that Def6 is a key bone remodeling regulator that regulates both osteoclasts and osteoblasts to maintain bone homeostasis. In disease settings, Def6 deficient DO11.10 mice on the Balb/c background with TCR activation developed a RA-like joint disease with bone erosion (*Chen et al., 2008*). In our previous study, we uncovered a key role of Def6 in inhibiting TNF-induced osteoclast formation and inflammatory bone resorption, and importantly, we provided evidence demonstrating a significant reverse correlation between Def6 expression levels and TNF-α activity in RA osteoclast precursors (*Binder et al., 2017*). Thus, our results support an important role for Def6 in human RA disease. In addition to chronic inflammation, the bone remodeling in RA is deregulated, featured by enhanced bone resorption but drastically reduced new bone formation, which is a long-standing challenge for treating bone damage in RA. We found that TNFα drastically decreases Def6 expression in osteoclast differentiation, and the decreased Def6 in turn promotes osteoclastogenesis and bone resorption. However, TNFα does not significantly affect Def6 expression in osteoblasts (*Figure 1—figure supplement 5*). Thus, Def6 maintains its inhibitory role in osteogenesis in this inflammatory setting. The differential regulation of Def6 expression by TNFα in osteoclasts and osteoblasts may contribute to enhanced bone resorption and reduced bone formation in the presence of TNFα. Taken together, the Def6-IFNβ axis plays a key role in regulating the differentiation of both osteoclasts and osteoblasts to maintain bone homeostasis, and targeting Def6 in bone cells might represent an effective strategy to control inflammatory bone resorption.

## Materials and methods

### Key resources table

| Reagent type (species) or resource | Designation | Source or reference | Identifiers | Additional information |
|---|---|---|---|---|
| Genetic reagent (*M. musculus*) | Def6$^{-/-}$ | PMID:16470246 | RRID:MGI:3628445 | |
| Genetic reagent (*M. musculus*) | Ifnar$^{-/-}$ | PMID:8009221 | RRID:MGI:3765898 | |
| Cell line (*Homo sapiens*) | Platinum-E (Plat-E) | Cell Biolabs | Cat# RV-101 RRID:CVCL_B488 | Has been authenticated by STR profiling and tested negative for mycoplasma |
| Cell line (*M. musculus*) | ST2 | DSMZ | Cat# ACC333 RRID:CVCL_2205 | Has been authenticated by STR profiling and tested negative for mycoplasma |
| Cell line (*M. musculus*) | MC3T3-E1 | ATCC | Cat# CRL-2594 RRID:CVCL_5437 | Has been authenticated by STR profiling and tested negative for mycoplasma |
| Antibody | anti-IBP/Def6 (rabbit polyclonal) | PMID:12923183 | | WB (1:1000) IF (1:100) |
| Antibody | anti-IFN-α/βRα (mouse monoclonal) | Santa Cruz Biotechnology | Cat# sc-7391 RRID:AB_2122749 | WB (1:1000) |
| Antibody | anti-mouse IFN Beta, neutralizing (rabbit polyclonal) | PBL Assay Science | Cat# 32400–1 RRID:AB_387872 | 10 U/ml |
| Antibody | anti-Stat1 (rabbit polyclonal) | Santa Cruz Biotechnology | Cat# sc-346 RRID:AB_632435 | WB (1:1000) |
| Antibody | anti-Stat3 (rabbit monoclonal) | Cell Signaling Technology | Cat# 12640 RRID:AB_2629499 | WB (1:1000) |
| Antibody | anti- Phospho-Stat1 (Ser727) (rabbit polyclonal) | Cell Signaling Technology | Cat# 9177 RRID:AB_2197983 | WB (1:1000) |
| Antibody | anti- Phospho-Stat1 (Tyr701) (rabbit polyclonal) | Cell Signaling Technology | Cat# 9171 RRID:AB_331591 | WB (1:1000) |
| Antibody | anti-Phospho-Stat3 (Tyr705) (rabbit polyclonal) | Cell Signaling Technology | Cat# 9131 RRID:AB_331586 | WB (1:1000) |

*Continued on next page*

*Continued*

| Reagent type (species) or resource | Designation | Source or reference | Identifiers | Additional information |
|---|---|---|---|---|
| Antibody | anti-PKR (mouse monoclonal) | Santa Cruz Biotechnology | Cat# sc-6282 RRID:AB_628150 | WB (1:1000) |
| Antibody | anti-p38α (rabbit polyclonal) | Santa Cruz Biotechnology | Cat# sc-535 RRID:AB_632138 | WB (1:3000) |
| Antibody | anti-Gapdh (rabbit polyclonal) | Santa Cruz Biotechnology | Cat# sc-25778 RRID:AB_10167668 | WB (1:3000) |
| Recombinant DNA reagent | pMXs-IRES-GFP (plasmid) | Cell Biolabs | Cat# RTV-013 | |
| Recombinant DNA reagent | pMXs-mPKR-FLAG-IG (plasmid) | Addgene | Cat# 101792 RRID:Addgene_101792 | |
| Sequence-based reagent | siEif2ak2 LNA | Qiagen | Cat#300600 | 40 nM for knockdown |
| Peptide, recombinant protein | Recombinant mouse IFNβ | PBL Assay Science | Cat#12400–1 | 10 U/ml |
| Peptide, recombinant protein | Recombinant Murine TNFα | Peprotech | Cat# 315-01A | 40 ng/ml |
| Peptide, recombinant protein | Recombinant Murine M-CSF | Peprotech | Cat# 315–02 | 20 ng/ml |
| Commercial assay or kit | VeriKine-HS Mouse IFN Beta Serum ELISA Kit | PBL Assay Science | Cat#42410–1 | |
| Commercial assay or kit | Mouse IFN Alpha All Subtype ELISA Kit | PBL Assay Science | Cat#42115–1 | |
| Chemical compound, drug | FuGENE 6 Transfection Reagent | Promega | Cat# E2691 | |
| Chemical compound, drug | TransIT-TKO Transfection Reagent | Mirus | Cat#MIR2150 | |
| Software, algorithm | HISAT2 | PMID:31375807 | RRID:SCR_015530 | http://daehwankimlab.github.io/hisat2/ |
| Software, algorithm | HTseq | PMID:25260700 | RRID:SCR_005514 | https://htseq.readthedocs.io/en/master |
| Software, algorithm | edgeR | PMID:19910308 PMID:22287627 | RRID:SCR_012802 | https://bioconductor.org/packages/ release/bioc/html/edgeR.html |
| Software, algorithm | pheatmap | https://cran.r-project.org/web/packages/ pheatmap/index.html | RRID:SCR_016418 | https://cran.r-project.org/web/packages /pheatmap/index.html |
| Software, algorithm | Enrichr | PMID:23586463 PMID:27141961 | RRID:SCR_001575 | https://maayanlab.cloud/Enrichr/ |
| Software, algorithm | ImageJ (v1.50i) | PMID:22930834 | RRID:SCR_003070 | https://imagej.nih.gov/ij/index.html |
| Software, algorithm | Graphpad Prism 8 | GraphPad Software | RRID:SCR_002798 | |
| Software, algorithm | Osteomeasure | OsteoMetrics, Inc | | |
| Other | ProLong Gold Antifade Mountant with DAPI | Thermo Fisher scientific | Cat#P36941 | |

## Mice and analysis of bone phenotype

*Def6⁻/⁻* mice have been described previously (*Fanzo et al., 2006*). *Def6⁻/⁻* mice used in this study have been backcrossed with C57/BL6 mice for more than 10 times, and do not develop autoantibodies or spontaneous autoimmunity disease. We thus took advantage of the *Def6⁻/⁻* mice on the C57/BL6 background to study bone biology to avoid the complex impact of the autoimmune disease severity and pattern that occur in other mouse strains. *Def6⁻/⁻* newborn (0-d3 after birth) mice and the wild type (WT) littermate controls were used for in vitro experiments. Eight week old *Def6⁻/⁻* mice and littermate WT mice were used for the in vivo experiments. *Ifnar1⁻/⁻* mice (IFNαβR KO mice, JAX stock number 010830 and MMRRC stock #32045) were a gift from Dr. Shuibing Chen (Weill Cornell Medicine). All mouse experiments were approved by Institutional Animal Care and Use Committee of the Hospital for Special Surgery and Weill Cornell Medical College.

For dynamic histomorphometric measures of bone formation, calcein (25 mg/kg, Sigma) was injected into mice intraperitoneally at 5 and 2 days before sacrifice to obtain double labeling of newly formed bones. The non-decalcified tibia bones were embedded in methyl methacrylate. 5 µm thick sections were sliced using a microtome (Leica RM2255, Leica Microsystems, Germany). For static histomorphometric measures of osteoblast parameters, non-decalcified sections of the tibiae were stained using Masson-Goldner staining kit (MilliporeSigma). The Osteomeasure software was used for bone histomorphometry using standard procedures according to the program's instruction.

## Reagents

Murine TNF-α and M-CSF were purchased from Peprotech. Recombinant mouse IFN-β was from PBL Assay Science. The control IgG (rabbit) was obtained from Santa Cruz Biotechnology, and IFN-β neutralizing antibody (rabbit polyclonal antibody against mouse interferon-beta) was from PBL Assay Science. Mouse IFN Alpha All Subtype ELISA Kit and VeriKine-HS Mouse IFN Beta ELISA Kit were purchased from PBL Assay Science. MojoSort Mouse CD45 Nanobeads were obtained from Biolegend.

## Cell culture

In vitro osteoblast differentiation: primary osteoblastic cells were isolated from the calvaria of newborn (0-3d) mice by enzymatic digestion in α-MEM with 0.1% collagenase (Worthington) and 0.2% dispase (Gibco) as described (*Ogata et al., 2000*), and were cultured in α-MEM medium (ascorbic acid free, Gibco) with 10% FBS (Atlanta Biologicals) and penicillin-streptomycin (Gibco) to expand for 6 days. The calvarial osteoblasts, ST2 or MC3T3-E1 osteoblasts were plated (6 × 10$^4$ cells per well in a 24-well dish pre-coated with 0.1% Gelatin solution (ATCC)) and cultured in osteogenic medium containing α-MEM with 10% FBS supplemented with 100 µg/ml ascorbic acid (Sigma) and 10 mM β-glycerophosphate (Sigma) to induce osteoblast differentiation. The osteogenic media were exchanged every 3 days. After 7 or 9 d, ALP staining was performed as described (*Zhao et al., 2006*), and after 14 or 21 d, bone nodules formed and were stained by alizarin red staining as described (*Gregory et al., 2004*). Extracted alizarin red solution in 10% acetic acid was measured by the absorbance at 405 nm.

Bone marrow harvested from WT and *Def6*$^{-/-}$ littermates was cultured in α-MEM with 10% FBS, penicillin-streptomycin and 2.4 mM L-glutamine (Gibco) in the presence of murine M-CSF (20 ng/ml) for 3 days to induce bone marrow derived macrophages (BMMs). BMMs were then replated and co-cultured in the upper chamber (2 × 10$^4$ cells per well in a 96-well transwell insert (Corning)) with calvarial osteoblasts plated in the bottom chamber (6 × 10$^4$ cells per well in a 24-well dish pre-coated with 0.1% Gelatin solution) in osteogenic medium as described above.

## Cell lines: The platinum-E (Plat-E), ST2, and MC3T3-E1 cell lines have been authenticated by STR profiling and tested negative for mycoplasma

### Retroviral gene transduction

Retrovirus packaging was performed by transfecting the retroviral vectors pMX-control (Cell Biolabs) and pMX-mouse PKR (Addgene) into Plat-E cells (Cell Biolabs) using FuGENE6 (Promega), as reported previously (*Zhao et al., 2009*). Calvarial osteoblasts were infected with the retrovirus with 8 µg/ml polybrene for 24 hr. The media was changed to ascorbic acid free media for at least 5 hr before stimulating with osteogenic media.

## Reverse transcription and real-time PCR

Reverse transcription and real-time PCR were performed as previously described (*Li et al., 2014*). The primers for real-time PCR were as follows: *Alpl*: 5'- CTTGACTGTGGTTACTGCTG -3' and 5'- CTTGACTGTGGTTACTGCTG -3'; *Bsp*: 5'- AATGGAGACGGCGATAGTTCCG -3' and 5'- GGAAAGTGTGGAGTTCTCTGCC -3'; *Bglap*: 5'- GCAATAAGGTAGTGAACAGACTCC -3' and 5'- CCATAGATGCGTTTGTAGGCGG -3'; *Runx2*: 5'- taagaagagccaggcaggtg-3' and 5'- tagtgcattcgtgggttgg-3'; *Eif2ak2*: 5'-AACCCGGTGCCTCTTTATTC-3' and 5'-ACTCCGGTCACGATTTGTTC-3'; *Mx1*: 5'-GGCAGACACCACATACAACC-3' and 5'- CCTCAGGCTAGATGGCAAG-3'; *Ifit1*: 5'-CTCCACTTTCAGAGCCTTCG-3' and 5'-TGCTGAGATGGACTGTGAGG-3'; *Ifit2*: 5'- AAATGTCATGGGGTACTGGAG

TT -3' and 5'- ATGGCAATTATCAAGTTTGTGG -3'; *Stat1*: 5'- CAGATATTATTCGCAACTACAA -3' and 5'- TGGGGTACAGATACTTCAGG -3'; *Cxcl10*: 5'- ATTCTTTAAGGGCTGGTCTGA-3' and 5'-CACCTCCACATAGCTTACAGT-3'; *Ifnb:* 5'-ttacactgcctttgccatcc-3' and 5'-agaaacactgtctgctggtg-3'; *Irf7:* 5'-CAGCGAGTGCTGTTTGGAGAC-3' and 5'-AAGTTCGTACACCTTATGCGG-3'; *Ifnar1:* 5'-acctgtgtcatgtgtgcttc-3' and 5'-tgaagcatctttccgtgtgc-3'; *Ifnar2:* 5'-agataagtggttggagggcatg -3' and 5'-tcaaattctggcggctcaag-3'; *Gapdh*: 5'-ATCAAGAAGGTGGTGAAGCA-3' and 5'-AGACAACCTGGTCCTCAGTGT-3'.

## Immunoblot analysis

Total cell extracts were obtained using lysis buffer containing 20 mM HEPES (pH 7.0), 300 mM NaCl, 10 mM KCl, 1 mM MgCl$_2$, 0.1% Triton X-100, 0.5 mM DTT, 20% glycerol, and 1 x proteinase inhibitor cocktail (Roche). The cell membrane-permeable protease inhibitor Pefabloc (1 mM, Sigma-Aldrich) was added immediately before harvesting cells. The protein concentration of extracts was quantified using the BCA protein assay kit (Pierce). Cell lysates (10 μg/sample) were fractionated on 7.5% SDS-PAGE, transferred to Immobilon-P membranes (Millipore) and incubated with specific antibodies. Western Lightning plus-ECL (PerkinElmer) was used for detection. Densitometry was performed using ImageJ software (National Institutes of Health). Def6 antibody was produced by the Dr. Pernis lab (*Gupta et al., 2003*). p-STAT1 (Y701), p-STAT1 (S727), p-STAT3 (Y705) and STAT3 antibodies were purchased from Cell Signaling, and STAT1 antibody was purchased from Santa Cruz. Type-I interferon receptor (IFNαR), PKR, p38α, and Gapdh antibodies were from Santa Cruz. ImageJ was used to quantify the immunoblot band density that reflects the amount of proteins.

## Immunofluorescence staining

Femoral bones from WT mice and *Def6*$^{-/-}$ mice were collected and immediately fixed in 4% paraformaldehyde solution for overnight. Bones were decalcified with 0.5 M EDTA at 4°C. All samples were embedded in paraffin and sliced into 6-um-thick sections. After deparaffinized and hydrated, sections were treated with 0.1% trypsin for 30 min at 37°C. Subsequently, sections were blocked with 1% BSA in PBST at room temperature for 60 min and incubated overnight at 4°C with anti-Def6 antibody (1:100). Sections were incubated with anti-rabbit secondary antibody conjugated with Alexa Fluor 700 (1:500, Cell signaling). Nuclei were counterstained with DAPI.

## In vitro gene silencing

Antisense inhibition using locked nucleic acid (LNA) technology from QIAGEN was applied to silence gene expression in vitro. LNA oligonucleotides specifically targeting *Eif2ak2* (PKR-LNA) and non-targeting control LNAs (Control-PKR) were from QIAGEN and were transfected into murine ST2 cells at concentrations of 40 nM using TransIT-TKO transfection reagent (Mirus) in accordance with the manufacturer's instructions.

## RNA-seq and Bioinformatics analysis

Total RNA was extracted using RNeasy Mini Kit (QIAGEN) following the manufacturer's instructions. NEBNext Ultra II RNA Library Prep Kit for Illumina (NEB) was used to purify poly-A+ transcripts and generate libraries with multiplexed barcode adaptors following the manufacturer's instructions. All samples passed quality control analysis using a Bioanalyzer 2100 (Agilent). High-throughput sequencing was performed using the Illumina HiSeq 4000 in the Weill Cornell Medical College Genomics Resources Core Facility. RNA-seq reads were aligned to the mouse genome (mm10) using HISAT2. HTseq was subsequently used to count reads in features and then edgeR was used to estimate the transcript abundances as CPM (counts per million) values. Genes with low expression levels (<1 cpm) in all conditions were filtered from downstream analyses. Benjamini-Hochberg false discovery rate (FDR) procedure was used to correct for multiple testing. Genes with p<0.05 were identified as significantly differentially expressed genes (DEG) between conditions using the edgeR analysis of two RNA-seq biological replicates. Volcano plot was generated by ggplot2 package in R. Heatmaps were generated by the pheatmap package in R. Pathway analysis was performed using WikiPathways in Enrichr (*Chen et al., 2013*; *Kuleshov et al., 2016*), input with the genes that were more highly expressed in *Def6*$^{-/-}$ osteoblasts than WT osteoblasts (≥1.2 fold, p<0.05) or more highly expressed in WT osteoblasts than *Def6*$^{-/-}$ osteoblasts (≥1.2 fold, p<0.05). Enriched pathways were ranked

based on the Combined Score calculated by the software. RNA-seq data (accession #GSE148455) have been deposited in NCBI's Gene Expression Omnibus (http://www.ncbi.nlm.nih.gov/geo/query/acc.cgi?acc=GSE148455).

## Statistical analysis

Statistical analysis was performed using Graphpad Prism software. Sample sizes were calculated on the assumption that a 20% difference in the parameters measured would be considered biologically significant with an estimate of sigma of 20% of the expected mean ($\alpha$ set at 0.05). For all in vivo experiments, at least five mice per genotype were used in each group, which provided a power of 0.92 to detect 20% differences in our studies ($\alpha$ set at 0.05). Student's $t$-test was applied if there are only two groups of samples. In the case of more than two groups of samples with one condition, one-way ANOVA followed by Tukey's post hoc test was used to calculate difference between any groups of samples. In the case of more than 2 groups of samples with more than one condition/treatment, two-way ANOVA followed by Sidak's multiple comparisons were used. $p$ value < 0.05 was taken as statistically significant; *$p$ value < 0.05 and **$p$ value < 0.01.

## Acknowledgements

We are grateful to Drs. Jian Luo and Ren Xu for their helpful discussions and assistance with histology. M.B.G. holds award DP5OD021351 and R01 AR075585 from the NIH, a Career Award for Medical Scientists from the Burroughs Welcome Foundation, and a Pershing Square Sohn Prize for Young Investigators in Cancer Research. This work was supported by grants from the National Institutes of Health (AR062047, AR068970 and AR071463 to BZ). The content of this manuscript is solely the responsibilities of the authors and does not necessarily represent the official views of the NIH.

## Additional information

### Funding

| Funder | Grant reference number | Author |
|---|---|---|
| NIAMS | AR062047 | Baohong Zhao |
| NIH | DP5OD021351 | Matthew Greenblatt |
| Burroughs Wellcome Fund | a Career Award for Medical Scientists | Matthew Greenblatt |
| Pershing Square Sohn Prize | Young Investigators in Cancer Research | Matthew Greenblatt |
| NIAMS | AR068970 | Baohong Zhao |
| NIAMS | AR071463 | Baohong Zhao |
| NIH | AR075585 | Matthew Greenblatt |

The funders had no role in study design, data collection and interpretation, or the decision to submit the work for publication.

### Author contributions

Zhonghao Deng, Data curation, Formal analysis, Validation, Investigation, Methodology, Writing - review and editing; Courtney Ng, Formal analysis, Validation, Investigation, Methodology, Writing - review and editing; Kazuki Inoue, Software, Formal analysis, Investigation, Methodology, Writing - review and editing; Ziyu Chen, Formal analysis, Investigation; Yuhan Xia, Investigation, Visualization; Xiaoyu Hu, Visualization, Methodology, Writing - review and editing; Matthew Greenblatt, Resources, Methodology, Writing - review and editing; Alessandra Pernis, Resources, Visualization, Writing - review and editing; Baohong Zhao, Conceptualization, Resources, Data curation, Supervision, Funding acquisition, Writing - original draft, Project administration, Writing - review and editing

## Author ORCIDs
Kazuki Inoue (ID) https://orcid.org/0000-0001-6305-9374
Baohong Zhao (ID) https://orcid.org/0000-0002-1286-0919

## Ethics
Animal experimentation: All mouse experiments were approved by Institutional Animal Care and Use Committee of the Hospital for Special Surgery and Weill Cornell Medical College (IACUC protocol #2016-0004).

## Decision letter and Author response
Decision letter https://doi.org/10.7554/eLife.59659.sa1
Author response https://doi.org/10.7554/eLife.59659.sa2

# Additional files

## Supplementary files
• Transparent reporting form

## Data availability
RNA-seq data (accession #GSE148455) have been deposited in NCBI's Gene Expression Omnibus (http://www.ncbi.nlm.nih.gov/geo/query/acc.cgi?acc=GSE148455).

The following datasets were generated:

| Author(s) | Year | Dataset title | Dataset URL | Database and Identifier |
|---|---|---|---|---|
| Zhao B, Deng Z, Ng C, Inoue K, Chen Z, Xia Y, Hu X, Greenblatt M, Pernis A | 2020 | Def6 regulates endogenous type I interferon responses in osteoblasts and suppresses osteogenesis | http://dx.doi.org/10.7554/dryad.s1rn8pk53 | Dryad Digital Repository, 10.5061/dryad.s1rn8pk53 |
| Zhao B, Deng Z, Ng C, Inoue K, Chen Z, Xia Y, Hu X, Greenblatt M, Pernis A | 2020 | Def6 regulates endogenous type-I interferon responses in osteoblasts and suppresses osteogenesis | http://www.ncbi.nlm.nih.gov/geo/query/acc.cgi?acc=GSE148455 | NCBI Gene Expression Omnibus, GSE148455 |

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
