## [Decision Letter]

**Acceptance summary:**

Your documentation that the newly identified molecule Def6 has a fundamental role in regulating bone formation through interferon-regulated genes adds a new dimension to our understanding of bone remodeling, besides unraveling a new therapeutic target. The studies were noted to be well done, and the review comments were addressed satisfactorily.

**Decision letter after peer review:**

Thank you for submitting your article "Def6 regulates endogenous type I interferon responses in osteoblasts and suppresses osteogenesis" for consideration by *eLife*. Your article has been reviewed by three peer reviewers, and the evaluation has been overseen by a Reviewing Editor and Clifford Rosen as the Senior Editor. The reviewers have opted to remain anonymous.

The reviewers have discussed the reviews with one another and the Reviewing Editor has drafted this decision to help you prepare a revised submission.

As the editors have judged that your manuscript is of interest, but as described below that additional experiments are required before it is published, we would like to draw your attention to changes in our revision policy that we have made in response to COVID-19 (https://elifesciences.org/articles/57162). First, because many researchers have temporarily lost access to the labs, we will give you 6 months to submit revised manuscripts. We are also offering, if you choose, to post the manuscript to bioRxiv (if it is not already there) along with this decision letter and a formal designation that the manuscript is "in revision at *eLife*". Please let us know if you would like to pursue this option. (If your work is more suitable for medRxiv, you will need to post the preprint yourself, as the mechanisms for us to do so are still in development.)

Summary:

The authors show that the Def6 deletion in mice resulted in attenuation of type I IFN signaling and enhanced osteoblastogenesis in vitro and in vivo. They propose that TNF-α suppressed osteoblastogenesis through Def6. This is a well-written manuscript and the main conclusion is novel and well supported by comprehensive in vitro and in vivo studies. The interest of the role of Def6 in bone remodeling is highly relevant since DEF6 locus in humans has been associated with bone mineral density. Despite these and other merits, there are a number of concerns that need to be addressed in order to improve the presentation. Most notably, the results are not consistent with previous reports showing that type I IFN signaling does not affect osteoblastogenesis (as seen in IFNAR1^-/-^ and *Stat1^-/-^* mice). The information about type I IFN expressing cells is not shown. Equally importantly, the mechanism of TNF-Def6-IFN-osteoblast relation is not well explored. These two latter points must be fully addressed in the revision. Other comments are noted below.

Essential revisions:

1) A major concern is the lack of proof that the observed effects are due primarily to expression of Def6 in osteoblasts and not a consequence of Def6 deletion in osteoclasts. There are no data showing convincingly that Def6 is expressed in osteoblasts. The mouse calvarial cell cultures used are not pure osteoblast cultures but contain osteoclast progenitor cells. Given the fact that osteoclasts can release factors regulating osteoblast activity, it might be that the observed expression of Def6 in these osteoblast cultures is mainly (or totally) reflecting expression in osteoclasts and that the effects on osteoblast differentiation may be due to regulation of coupling factors by Def6 in osteoclasts. The same might be true also for the observations made in the mice with global deletion of Def6, particularly as these mice develop osteoporosis due to enhanced osteoclastogenesis (Binder et al., 2017). To clarify the importance of Def6 in osteoblast differentiation in vivo, it would be desirable to analyze osteoblast-specific Def6 conditional knockout mice. However, while there is review consensus that this may not be possible in the short period, in vitro experiments are required to convincingly prove this. For example, the authors could CRISPR Def6 in osteoclasts and osteoblasts in dual chamber cultures.

2) To support the authors' conclusion, osteoblasts should express functional type I IFN receptors. The authors should show type I IFN receptor expression in osteoblasts of wild-type and the *Def6^-/-^* mice.

3) The *Def6^-/-^* mice have an osteoporotic phenotype in a resting state. If the IFN signaling pathway is the main stream for the high bone turnover of the mice, endogenous type I interferon should be important for WT mice. Which cells produce the type I IFN to maintain normal bone remodeling?

4) The authors considered only IFN-b in this manuscript. Since type I IFN signaling is driven also by IFN-a, a contribution of IFN-a should be investigated. For example, the authors should analyze IFN-a expression in vivo and effects of IFN-a antibody/IFN-a treatment in osteoblastogenesis.

5) The authors concluded that the *Def6^-/-^* mice exhibit an osteoporotic phenotype due to the impairment of the type I IFN signaling based on the phosphorylation of Stat1. However, it has been reported that *Stat1^-/-^* mice have a high bone mass phenotype, and their accelerated osteoblastogenesis was independent of type I IFN signaling (Kim et al., 2003). Furthermore, there is a report showing that IFNAR1^-/-^ calvarial cells have normal osteoblast differentiation in vitro (Takayanagi et al., 2002). The authors should provide data to explain the discrepancy.

6) The authors only showed the expression of IFN-responsive genes and Stat1 phosphorylation in *Def6^-/-^* cells. It remains unclear how Def6 regulates the IFN signaling pathway in osteoblasts. The authors should show a mechanism by which Def6 activates the IFN signaling.

7) The authors should show the expression of IFN-b after TNF-α stimulation in WT and *Def6^-/-^* mice. The authors should show Def6 expression after TNF-α stimulation in WT mice.

8) If TNF-α effect is Def6 dependent, why TNF-α completely suppressed osteoblastogenesis in *Def6^-/-^* mice?

9) It is clear from the data that the potent inhibitory effect of TNF-α on osteogenesis is independent of Def6, but the paragraph relating to Figure 6 was constructed in a way as if TNF-α is actually doing something in osteoblasts through Def6 (it was actually shown that way in Figure 6D). It is unclear whether Figure 6 adds anything to the manuscript. If the authors would like to keep the figure, the Results section will need to be written in a much more straightforward way to avoid any confusion.

---

## [Author Response]

Essential revisions:1) A major concern is the lack of proof that the observed effects are due primarily to expression of Def6 in osteoblasts and not a consequence of Def6 deletion in osteoclasts. There are no data showing convincingly that Def6 is expressed in osteoblasts. The mouse calvarial cell cultures used are not pure osteoblast cultures but contain osteoclast progenitor cells. Given the fact that osteoclasts can release factors regulating osteoblast activity, it might be that the observed expression of Def6 in these osteoblast cultures is mainly (or totally) reflecting expression in osteoclasts and that the effects on osteoblast differentiation may be due to regulation of coupling factors by Def6 in osteoclasts. The same might be true also for the observations made in the mice with global deletion of Def6, particularly as these mice develop osteoporosis due to enhanced osteoclastogenesis (Binder et al., 2017). To clarify the importance of Def6 in osteoblast differentiation in vivo, it would be desirable to analyze osteoblast-specific Def6 conditional knockout mice. However, while there is review consensus that this may not be possible in the short period, in vitro experiments are required to convincingly prove this. For example, the authors could CRISPR Def6 in osteoclasts and osteoblasts in dual chamber cultures.

We showed Def6 expression in calvarial osteoblasts in Figure 1A and B using both qPCR and western blot approaches in the original submission. Following the reviewers' comments, we further confirmed Def6 expression in osteoblasts in vivo on the bone slices using immunofluorescence staining approach with Def6 KO mice as a negative control in Figure 1—figure supplement 1A. We also found Def6 expression in pure osteoblastic cell lines, such as ST2 and MC3T3 cells (Figure 1—figure supplement 1B, C). These results provide solid data showing Def6 expression in osteoblasts. We appreciate the review panel's consensus that generation of osteoblast-specific Def6 conditional knockout mice is not possible in a short period, in particular during the COVID-19 pandemic. While using calvarial osteoblasts is a well-established and widely used method for in vitro examination of osteoblast differentiation and mechanistic studies, we agree that this culture system is a relatively pure, but not an absolutely pure osteoblast culture. In fact, most of current in vitro primary cell cultures are not able to reach an absolutely pure level. We understand the reviewers' concern, and we have performed a series of experiments to extensively verify whether the calvarial osteoblast culture system is solid to investigate the role of Def6 in osteoblastogenesis without significant macrophage lineage effects. First, using RNAseq, a sensitive approach, we can only detect very low expression levels (below 1-2 counts per million reads) of marker genes for either macrophages or osteoclasts in this culture system, indicating a very low level of macrophage lineage cells in the culture system. Furthermore, these genes belong to non-significant genes and there are no difference in their expression between WT and Def6 deficient osteoblast cultures (Figure 1—figure supplement 2). Secondly, to exclude any potential effect from hematopoietic cells, we removed CD45 positive cells, which label all hematopoietic cells including macrophages/osteoclast progenitors, and selected CD45 negative calvarial osteoblasts to compare the osteoblast differentiation. Similarly as the results in unselect cultures shown in Figures 1 and 3, Def6 deficiency significantly enhanced osteoblastogenesis and osteoblast marker gene expression, but reduced ISG gene expression without CD45 cells in cultures (Figure 1—figure supplement 3). Thirdly, we co-cultured CD45 negative calvarial osteoblasts (without any hematopoietic cells) with bone marrow derived macrophages (BMMs) isolated from WT or Def6 deficient mice in a dual chamber. We found no different effect from Def6 deficiency in BMMs on osteoblast differentiation (Figure 1—figure supplement 4). Therefore, these results collectively validated this well-established osteoblast culture system, which is solid, feasible and convincing to test Def6 function in osteoblast differentiation. These new results are now shown in Figure 1—figure supplement 1, 2, 3, 4 and noted in the Results, Discussion and figure legends.

2) To support the authors' conclusion, osteoblasts should express functional type I IFN receptors. The authors should show type I IFN receptor expression in osteoblasts of wild-type and the Def6^-/-^ mice.

Following the reviewers' suggestion, we have detected the expression of type I IFN receptors at both protein and mRNA levels in osteoblasts of WT and Def6 KO mice (new Figure 5A and Figure 5—figure supplement 1). Def6 deficiency does not significantly affect the expression levels of type I IFN receptors in osteoblasts. Furthermore, these receptors are functional evidenced by the downstream activation of STAT1 and STAT3 in response to IFNβ as well as ISG gene expression (Figure 5B, C and Figure 3E). These results are now shown in Figure 5A, B, C, Figure 3E and Figure 5—figure supplement 1, and noted in the Results and figure legends.

3) The Def6^-/-^ mice have an osteoporotic phenotype in a resting state. If the IFN signaling pathway is the main stream for the high bone turnover of the mice, endogenous type I interferon should be important for WT mice. Which cells produce the type I IFN to maintain normal bone remodeling?

We tested serum IFNβ and IFNα levels in both WT and Def6 KO mice. IFNα level is nearly undetectable in the serum from WT or *Def6^-/-^* mice, and undetectable in CD45 negative osteoblast culture medium (Figure 6—figure supplement 2 and data not shown). In contrast, we detected IFNβ in WT mice, and the serum IFNβ level is significantly lower in *Def6^-/-^* mice than WT mice (new Figure 6A). Importantly, Def6 deficiency significantly decreased IFNβ level in CD45 negative osteoblast cells (Figure 6—figure supplement 1) as well as in osteoblast culture medium (new Figure 6B). These results demonstrate that osteoblasts produce IFNβ, and Def6 deficiency downregulates the endogenous IFNβ level secreted by osteoblasts. We are aware that other cell types, such as macrophages, osteoclasts and T cells, presumably produce IFNβ and thus may contribute to in vivo IFNβ level. Our evidence in this study supports that osteoblasts produce IFNβ which plays a significant feedback inhibitory role in osteoblastogenesis. These new results are now shown in new Figure 6A, B, Figure 6—figure supplement 1, 2, and noted in the Results, Discussion and figure legends.

4) The authors considered only IFN-b in this manuscript. Since type I IFN signaling is driven also by IFN-a, a contribution of IFN-a should be investigated. For example, the authors should analyze IFN-a expression in vivo and effects of IFN-a antibody/IFN-a treatment in osteoblastogenesis.

Following the reviewers' suggestion, we have examined IFNα levels in both WT and Def6 KO mice. IFNα level is nearly undetectable in the serum from WT or *Def6^-/-^* mice, and completely undetectable in CD45 negative osteoblast culture medium (Figure 6—figure supplement 2 and data not shown). IFNα level is also totally undetectable in osteoblastic cell lines, such as ST2 and MC3T3 cells (data not shown). Thus, there is no need in this study to test the effects of IFNα antibody/IFNα treatment on osteoblastogenesis. These new results are now shown in Figure 6—figure supplement 2, and noted in the Results and figure legend.

5) The authors concluded that the Def6^-/-^ mice exhibit an osteoporotic phenotype due to the impairment of the type I IFN signaling based on the phosphorylation of Stat1. However, it has been reported that Stat1^-/-^ mice have a high bone mass phenotype, and their accelerated osteoblastogenesis was independent of type I IFN signaling (Kim et al., 2003). Furthermore, there is a report showing that IFNAR1^-/-^ calvarial cells have normal osteoblast differentiation in vitro (Takayanagi et al., 2002). The authors should provide data to explain the discrepancy.

We acknowledge the contribution of these two publications to the field. On the other hand, however, we and other groups have shown that type I IFN signaling in osteoblasts activated by exogenous IFNβ inhibits osteoblastogenesis (Figure 5A, B, C, E, F, and Woeckel et al., 2012a and b; Kim et al., 2003). Our ELISA results clearly demonstrate that osteoblasts express and secrete IFNβ (Figure 6B, Figure 6—figure supplement 1). Blocking endogenous IFNβ activity via IFNβ blocking antibody significantly enhances osteoblast differentiation (Figure 4). Following the reviewers' suggestion, we further examined the effect of endogenous IFNβ on osteoblast differentiation using *Ifnar1^-/-^* calvarial osteoblasts isolated from *Ifnar1^-/-^*mice (IFNαβR KO mice, JAX stock number 010830 and MMRRC stock #32045), which do not activate downstream type I IFN signaling.

As shown in Figure 6C-E, deficiency of type I IFN receptor significantly enhanced osteoblastogenesis, evidenced by the increases in ALP activity (Figure 6C), bone nodule formation (Figure 6D) and osteoblast marker gene expression, such as *Alpl* and *Bglap* (Figure 6E). As expected, the expression of ISG genes, such as *Mx1*, *Ifit1* and *Irf7*, was markedly induced in the WT osteoblast cultures, but abolished in *Ifnar1^-/-^* osteoblasts (Figure 6E). By these multiple approaches and multiple verification methods, including RNAseq, ELISA, cell staining, mineralization assay, osteoblast marker gene expression and ISG expression at multiple time points during osteoblast differentiation, these results collectively corroborate the presence of endogenous IFNβ and IFNβ response during osteoblastogenesis, and demonstrate that endogenous IFNβ plays an inhibitory role in osteoblast differentiation. While it is difficult to assume the exact reason for the discrepancy between our conclusion and that by Takayanagi et al., 2002, it is likely due to variation in in vitro osteoblast culture method, approaches, time points of analysis and/or interpretation in different labs, which also occurs with other studies between different groups in the research field.

Nonetheless, we believe that our results supported by multiple approaches and multiple verification methods are convincing. We have cited the Takayanagi et al., 2002 article and noted the variable reports between groups. Stat1 is downregulated by Def6 deficiency (Figure 5D). Thus, we assume that Stat1 could function as one of the regulators downstream of Def6 and contribute to osteoblastic inhibition. We have cited the Kim et al., 2013 article, and discussed this possibility in our Discussion. In the current study, we focus on the novel Def6 function in osteoblasts, instead of Stat1. The related results are shown in Figure 4, Figure 5, Figure 6B-E and Figure 6—figure supplement 1 and their figure legends, and noted in the Results and Discussion sections.

6) The authors only showed the expression of IFN-responsive genes and Stat1 phosphorylation in Def6^-/-^ cells. It remains unclear how Def6 regulates the IFN signaling pathway in osteoblasts. The authors should show a mechanism by which Def6 activates the IFN signaling.

During the revision, we demonstrated the presence of endogenous IFNβ produced by osteoblasts (new Figure 6B and Figure 6—figure supplement 1) and IFNβ response during osteoblastogenesis (Figures 3 and 5). How IFNβ is induced in osteoblastic cells was unknown. As Def6 deficiency significantly reduced IFNβ expression in osteoblasts, we set off to investigate how Def6 regulates IFNβ expression and its downstream IFNβ response. We found that loss of Def6 decreased PKR (gene name: *Eif2ak2*) expression in osteoblasts (Figure 3E, 6F). PKR is an ISG gene and also functions as an activator of IFN-β expression (29-33). We then knocked down PKR expression in ST2 cells (Figure 6G), an osteoblast cell line well-established for osteoblast differentiation and mechanistic studies. Similarly as calvarial osteoblasts, the expression of *Ifnb* and ISG genes was highly induced during osteoblast differentiation in ST2 cells (Figure 6H white bars). PKR deficiency resulted in a drastic decrease in the expression of *Ifnb* and ISG genes, such as *Mx1, Ifit1, Stat1 and Irf7* (Figure 6H black bars), indicating that PKR is a key activator for IFN-β and downstream ISG gene induction in osteoblasts. In contrast, PKR deficiency significantly promoted osteoblast differentiation (Figure 6I) and the expression of osteoblastic genes, such as *Runx2, Alpl, Bsp and Bglap* (Figure 6J). Taken together, these new findings indicate that PKR controls endogenous IFN-β production and response in osteoblasts, thereby contributing to the feedback inhibitory effects mediated by endogenous IFN-β. Def6 deficiency significantly downregulates PKR expression level (Figures 3E, 6F). Furthermore, overexpression of PKR in Def6 KO osteoblasts suppresses osteoblastic marker genes but enhances *Ifnb* and ISG expression (Figure 6K-L). These results indicate that Def6 modulates endogenous IFNβ activity, at least partially via PKR, during osteoblastogenesis. These new mechanistic results are now shown in new Figure 6B, F, G, H, I, J, K, L, and noted in the Results, Discussion and figure legends.

We will address question 7, 8 and 9 together because these questions are similar and all related to the data in old Figure 6.

7) The authors should show the expression of IFN-b after TNF-α stimulation in WT and Def6^-/-^ mice. The authors should show Def6 expression after TNF-α stimulation in WT mice.8) If TNF-α effect is Def6 dependent, why TNF-α completely suppressed osteoblastogenesis in Def6^-/-^ mice?9) It is clear from the data that the potent inhibitory effect of TNF-α on osteogenesis is independent of Def6, but the paragraph relating to Figure 6 was constructed in a way as if TNF-α is actually doing something in osteoblasts through Def6 (it was actually shown that way in Figure 6D). It is unclear whether Figure 6 adds anything to the manuscript. If the authors would like to keep the figure, the Results section will need to be written in a much more straightforward way to avoid any confusion.

We thank the reviewers for their insightful comments, and agree that the old Figure 6 did not add much to the manuscript. We thus have now removed the old Figure 6 results. We also agree with the reviewers' point that "the potent inhibitory effect of TNF-α on osteogenesis is independent of *Def6*" and this is presumably because TNF does not affect Def6 expression in osteoblasts (Figure 1—figure supplement 5). We feel the differential regulation of Def6 by TNF in osteoclasts and osteoblasts interesting, which may contribute to the unbalanced TNF-mediated bone remodeling. We therefore added a short discussion "We found that TNFα drastically decreases Def6 expression in osteoclast differentiation, and the decreased Def6 in turn promotes osteoclastogenesis and bone resorption. […] The differential regulation of Def6 expression by TNFα in otesoclasts and osteoblasts may contribute to enhanced bone resorption and reduced bone formation in the presence of TNFα."